# OUT-OF-DISTRIBUTION PREDICTION WITH INVARIANT RISK MINIMIZATION: THE LIMITATION AND AN EFFECTIVE FIX

## ABSTRACT

This work considers the out-of-distribution (OOD) prediction problem where (1) the training data are from multiple domains and (2) the test domain is unseen in the training. DNNs fail in OOD prediction because they are prone to pick up spurious correlations. Recently, Invariant Risk Minimization (IRM) is proposed to address this issue. Its effectiveness has been demonstrated in the colored MNIST experiment. Nevertheless, we find that the performance of IRM can be dramatically degraded under *strong* $\Lambda$ *spuriousness* – when the spurious correlation between the spurious features and the class label is strong due to the strong causal influence of their common cause, the domain label, on both of them (see Fig. 1). In this work, we try to answer the questions: why does IRM fail in the aforementioned setting? Why does IRM work for the original colored MNIST dataset? Then, we propose a simple and effective approach to fix the problem of IRM. We combine IRM with conditional distribution matching to avoid a specific type of spurious correlation under strong $\Lambda$ spuriousness. Empirically, we design a series of semi synthetic datasets – the colored MNIST plus, which exposes the problems of IRM and demonstrates the efficacy of the proposed method.

## 1 INTRODUCTION

Strong empirical results have demonstrated the efficacy of deep neural networks (DNNs) in a variety of areas including computer vision, natural language processing and speech recognition. However, such positive results overwhelmingly rely on the assumption that the training, validation and test data consist of independent identical samples of the same underlying distribution. In contrast, in the setting of out-of-distribution (OOD) prediction where (1) the training data is from multiple domains and (2) the test set has different distribution from the training, the performance of DNNs can be dramatically degraded. This is because DNNs are prone to pick up spurious correlations which do not hold beyond the training data distribution (Beery et al., 2018; Arjovsky et al., 2019). For example, when most camel pictures in a training set have a desert in the background, DNNs will pick up the spurious correlation between a desert and the class label, leading to failures when camel pictures come with different backgrounds in a test set. Therefore, OOD prediction remains an extremely challenging problem for DNNs.

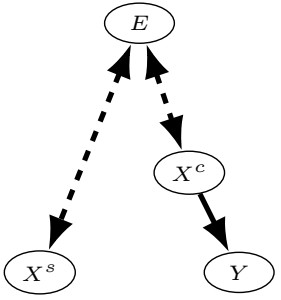

Figure 1: The causal graph in OOD prediction: $P(Y|X^c)$ is invariant across domains. The spurious correlation $P(Y|X^s)$ may vary. A directed (bidirected) edge is a causal relationship (correlation).

The invariant causal relationships across different domains turns out to be the key to address the challenge of OOD prediction. The causal graph in Fig. 1 describes the relationships of the variables in the OOD prediction problem. Although spurious correlations learned in one domain are unreliable in another, invariant causal relationships enable DNNs that capture causal relationships to generalize to unseen domains. In practice, it is extremely difficult to know whether an input feature is causal or spurious. Thus, a recipe for training DNNs that capture causal relationships is learning causal feature representations that hold invariant causal relationships with the class label.

Thus, the main challenge of OOD prediction becomes learning causal feature representations given training data from multiple domains. Invariant Risk Minimization (IRM) (Arjovsky et al., 2019) is proposed to learn causal feature representations for OOD prediction. IRM formulates the invariance of causal relationships as a constraint. It requires that causal feature representations must result in the same optimal classifier across all domains. IRM is written as the conditional independence, $Y \perp\!\!\!\perp E | F(X)$ in (Chang et al., 2020; Zeng et al., 2019) which can be derived from the causal graph in Fig. 1 when $F(X)$ is a mapping of causal features ($X^c$ in Fig. 1) with no information loss. A detailed discussion on IRM and $Y \perp\!\!\!\perp E | F(X)$ is in Appendix A.

Despite IRM's success in the colored MNIST dataset (Arjovsky et al., 2019), in this work, we find an important issue of IRM that has not been discussed. Specifically, we consider the situation where strong spurious correlations among the spurious features, the class label and the domain label only hold for training data. We name this *strong $\Lambda$ spuriousness* using the shape of the structure among $X^s, E$ and $Y$ in Fig. 1. Under strong $\Lambda$ spuriousness, IRM regularized empirical risk can have low values with spurious feature representations that accurately predict the domain label. This is because, in this setting, picking up spurious features can achieve high accuracy in predicting both domain and class in the training set, but not in the test. However, the colored MNIST dataset cannot expose this issue because the strong similarity between the two training domains makes it difficult to pick up the weak spurious correlation between the domain label and the spurious features. To illustrate this problem, we design a new dataset – the colored MNIST plus. As shown in Fig. 2, in this dataset, under strong $\Lambda$ spuriousness, the performance of IRM models is significantly degraded.

Moreover, to resolve this issue of IRM, we propose an effective solution, which combines IRM with conditional distribution matching (CDM). The CDM constraint means that the representation distribution of instances from the same class should be invariant across domains. Theoretically, we show that (1) causal feature representations can satisfy CDM and IRM at the same time and (2) CDM can prevent DNNs from learning spurious feature representations that accurately predict the domain label. Empirically, on our newly introduced dataset, the proposed method achieves significant performance improvement over IRM under strong $\Lambda$ spuriousness.

## 2 PRELIMINARIES AND IRM

**Notations.** We use lowercase (e.g., $x$), uppercase (e.g., $X$) and calligraphic uppercase (e.g., $\mathcal{X}$) letters for values, random variables and spaces. We let $X \in \mathcal{X}$, $Y \in \mathcal{Y}$ and $E \in \mathcal{E}$ denote raw input, the class label and the domain label where $\mathcal{X}$, $\mathcal{Y}$ and $\mathcal{E}$ are the spaces of input, class labels and domains. A DNN model consists of a feature learning function $F$ and a classifier $G$. A feature learning function $F : \mathcal{X} \to \mathbb{R}^d$ maps raw input $X$ to its $d$-dimensional representations $F(X)$. A classifier $G : \mathbb{R}^d \to \mathcal{Y}$ maps a feature representation to a class label. We denote their parameters by $\boldsymbol{\theta}_F$ and $\boldsymbol{\theta}_G$, respectively. Let $\boldsymbol{\theta} = Concat(\boldsymbol{\theta}_F, \boldsymbol{\theta}_G)$ denote the concatenation of them.

A domain $e$ of $n_e$ instances is denoted by $D_e = \{x_i^e, y_i^e\}_{i=1}^{n_e}$. Let $\mathcal{E}_{tr}$ and $\mathcal{E}_{ts}$ denote the set of training and test domains, in OOD prediction, we have (1) $|\mathcal{E}_{tr}| > 1$ and (2) $\mathcal{E}_{tr} \cap \mathcal{E}_{ts} = \emptyset$.

**Problem Statement.** Given data from multiple training domains $\{D_e\}_{e \in \mathcal{E}_{tr}}$. We aim to predict the label $y_i^{e'}$ of each instance with features $x_i^{e'}$ from a test domain $\{x_i^{e'}, y_i^{e'}\}_{i=1}^{n_{e'}}, e' \in \mathcal{E}_{ts}$.

**Invariant Risk Minimization.** IRM (Arjovsky et al., 2019) is a recently proposed method to impose the causal inductive bias: the causal relationships between causal features and the label should be invariant across different domains. It not only aims to address the challenging OOD prediction problem, but also is a pioneer work that guides causal machine learning research towards the development of inductive bias imposing causal constraints. The effectiveness of IRM and its variants have been demonstrated across various areas including computer vision (Ahuja et al., 2020), natural language processing (Chang et al., 2020), CTR prediction (Zeng et al., 2019), reinforcement learning (Zhang et al., 2020a) and financial forecasting (Krueger et al., 2020). Arjovsky et al. (2019) propose the original formulation of IRM as a two-stage optimization problem:

$$\underset{\boldsymbol{\theta}_F, \boldsymbol{\theta}_G}{\arg\min} \sum_{e \in \mathcal{E}_{tr}} \mathbb{E}_{(x,y) \sim D_e}[R^e(G(F(x)), y)]$$

$$s.t. \ \boldsymbol{\theta}_G \in \underset{\boldsymbol{\theta}_G'}{\arg\min} \ R^e(G(F(x); \boldsymbol{\theta}_G'), y), \tag{1}$$

where $R^e$ denotes the loss function of domain $e$. Then, they show that the IRM constraint can be imposed by adding a regularizer into the loss function. It turns out the IRM constraint can be written as the conditional independence $Y \perp\!\!\!\perp E|F(X)$ (Chang et al., 2020; Zeng et al., 2019). This can be derived from the causal graph in Fig. 1. For brevity, we refer to $Y \perp\!\!\!\perp E|F(X)$ as the IRM constraint.

## 3 THE LIMITATION OF IRM

In this section, we find that IRM fails under strong $\Lambda$ spuriousness – when the spurious correlations among spurious features, the domain label and the class label are strong. To show it, we design a new dataset – the colored MNIST plus (CMNIST+) to expose this limitation of IRM. Then, we try to answer two crucial research questions: (1) why does IRM fail in CMNIST+? (2) Why does IRM work for the original colored MNIST (CMNIST) dataset?

### 3.1 WHY DOES IRM FAIL?

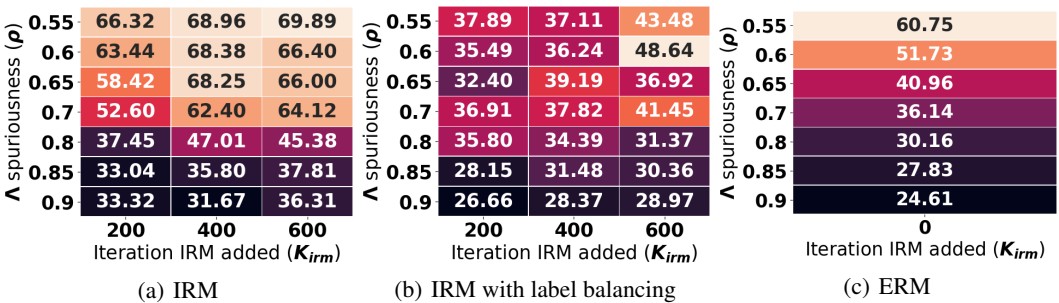

(a) IRM  (b) IRM with label balancing  (c) ERM

Figure 2: Test accuracy of IRM, IRM with balanced classes in each domain, and ERM on CM-NIST+: when the $\Lambda$ spuriousness is strong ($\rho \geq 0.8$), the performance of IRM drops dramatically because, in this situation, the spurious feature representation $F(X) = E$ satisfies the IRM constraint. The naïve solution, balancing classes in each domain cannot mitigate this problem of IRM.

Despite its previous success, results in Fig. 2 show that IRM fails in our newly designed colored MNIST plus dataset, under strong $\Lambda$ spuriousness ($\rho \geq 0.8$). Here, we show that, under strong $\Lambda$ spuriousness, at least two types of feature representations, $F(X) = Y$ and $F(X) = E$, satisfy the IRM constraint. When $F(X) = E$, the feature representations pick up spurious relationships.

**Satisfying IRM by Perfectly Predicting the Label.** First, we consider a case where there exists a feature representation that perfectly predicts the class label, i.e., $F(X) = Y$. In this case, we know that the feature representation satisfies the IRM constraint because $Y \perp\!\!\!\perp E|F(X) = Y$ always holds. This is reasonable because the causal relationship between $Y$ and itself must be invariant across different domains.

**Satisfying IRM by Perfectly Predicting the Domain Does Not Lead to OOD Generalizable Feature Representations.** However, there is another way to satisfy the IRM constraint. If there exists a feature representation $F(X) = E$, then the IRM constraint can be satisfied by $F(X)$. This is due to the fact $Y \perp\!\!\!\perp E|F(X) = E$. However, this is not desired behavior of IRM as $F(X) = E$ is a type of spurious feature representation. This is because $P(Y|E)$ can vary across domains. Using the camel picture example, if domains are the individuals who take the picture, we may find more camel pictures taken by person $e$ than from person $e'$ ($P(Y = camel|E = e) > P(Y = camel|E = e')$). Why is $F(X) = E$ a good solution for the original IRM optimization problem (Eq. 1)? This is because $F(X) = E$ also leads to low values of the loss function $R^e$ since the direct causal influence of the domain label $E$ on the class label $Y$ is also strong when $\Lambda$ spuriousness is strong. So, with a proper classifier $G$, we can achieve low values of $R^e$ with $F(X) = E$. Then, we describe the CMNIST+ dataset and show experimental and theoretical results to support our claim.

**The Colored MNIST Plus (CMNIST+) Dataset.** We follow CMNIST (Arjovsky et al., 2019) to create CMNIST+ by resampling and adding colors to instances of MNIST. The digits $0 - 4$ ($5 - 9$)

are class $Y = 1$ $(Y = 0)$. We randomly flip $25\%$ of class labels so that spuriousness can be stronger than the causal relationship between the shape $S$ (causal features) and the class label $Y$. Table 1 describes the dataset. The variable $C \in \{G, B, R\}$ denotes the color, which represents the spurious feature $X^s$. We explain why we use three colors for CMNIST+ in Appendix C.2. The parameter $\rho \in (0.5, 1)$ controls the strength of the spurious correlations between the color (spurious features) $C$ and the class label $Y$ through their common cause, the domain label $E$. The larger the value of $\rho$, the stronger the spurious correlations. Intuitively, in the two training domains of CMNIST+, in addition to the strong spurious correlation between the class label $Y$ and color $C$, we set the spurious correlation between the domain label $E$ and color $C$ to be strong, too. Thus, the CMNIST+ dataset can expose the problem of IRM: it would not penalize the models that fit the domain variable as the feature representation ($F(X) = E$). To verify this claim, we show analysis results to support the experimental results in Fig. 2. The data generating process of CMNIST+ can be found in Append C.2.

Table 1: Definition of CMNIST+

| $E$ | $P(Y = 1\|E)$ | $Y$ | $P(C = G\|Y, E)$ | $P(C = B\|Y, E)$ | $P(C = R\|Y, E)$ |
|---|---|---|---|---|---|
| $E = 1$ | 0.9 | $Y = 1$ | $\rho$ | $(1 - \rho)/2$ | $(1 - \rho)/2$ |
| | | $Y = 0$ | $(1 - \rho)/2$ | $(1 - \rho)/2$ | $\rho$ |
| $E = 2$ | 0.1 | $Y = 1$ | $(1 - \rho)/2$ | $\rho$ | $(1 - \rho)/2$ |
| | | $Y = 0$ | $(1 - \rho)/2$ | $(1 - \rho)/2$ | $\rho$ |
| $E = 3$ | 0.5 | $Y = 1$ | 0.1 | 0.1 | 0.8 |
| | | $Y = 0$ | 0.4 | 0.4 | 0.2 |

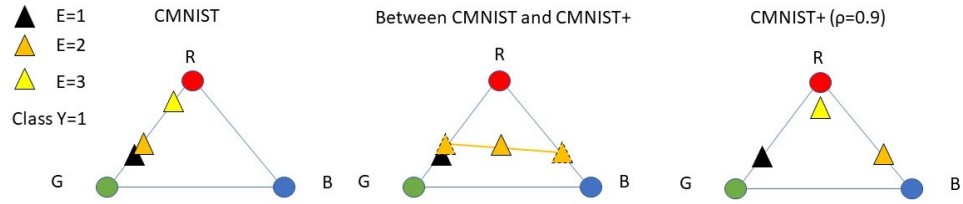

Figure 3: We visualize the differences between CMNIST and CMNIST+ ($\rho = 0.9$) in terms of $P(C|Y = 1, E)$. Each large triangle represents the space of $P(C|Y, E)$. Each small triangle shows the values of $P(C = c|Y = 1, E = e), c \in \{R, G, B\}, e \in \{1, 2, 3\}$.

**Analysis Results.** Here, we show the analysis results on the CMNIST+ to answer two questions: what features will be learned by ERM and IRM under under different $\Lambda$ spuriousness? What is the expected test accuracy of these models? For simplicity, we assume that the classifier is deterministic, which always predicts the majority class given the feature. If the number of instances from each class is the same, then it would predict a random label. We analyze three types of spurious feature representations: (1) those that fit color, (2) those that fit the domain label and (3) those that use a combination of them for prediction. The first case mimics the behavior of ERM that picks up the spurious correlation between color and the class label. The second one represents a model satisfying IRM by $F(X) = E$. The third one stands for a model satisfying IRM by $F(X) = Concat(E, C)$. One may argue that IRM can be fixed if we simply balance the two classes in each domain. Theoretical analysis shows that this is an invalid solution. We summarize results in Table 2 where $\hat{E}$ is the domain label predicted by a deterministic classifier using color as the feature. This is because in an unseen test domain, we cannot directly use the domain label for prediction. The highlighted numbers show which type of features would be learned by ERM and IRM. As $\rho$ increases, ERM and IRM are more likely to fit spurious features. This explains results in Fig. 2. As a DNN may still pick up some causal features (shape in CMNIST+) even when $\rho = 0.9$, the test accuracy of IRM in practice would be greater than $0.35$ without label balancing and $0.2$ with label balancing. Similarly, the test accuracy of ERM would be greater than $0.2$. From Fig. 2, we can see the test accuracy of IRM, IRM with label balancing and ERM are slightly better than the aforementioned lower bounds. The derivations can be found in Appendix B.

**Experimental Setup.** We use a LeNet-5 (LeCun et al., 1998) instead of a three-layer MLP. To take input with three colors, we modify the first CNN layer to have three channels. LeNet-5 has more predictive power such that it can pick up the three types of spurious correlations (color – domain, color – class, and domain – class) or the causal relationship (shape – class). We randomly split the instances from the training domains into $80\%$ training and $20\%$ validation. The model selection is done by picking the one with the lowest validation loss in each run. We report the average test accuracy of the selected models in 10 runs. It is crucial to ensure only data from training domains are used in model selection, since the test domain should be unseen during training and validation. During training, we begin applying the IRM penalty at iteration $K_{IRM}$, set to 200, 400, or 600. By varying $K_{IRM}$, we aim to examine the following hypothesis: IRM works by pushing the spurious features out of the representations learned by the standard ERM training before the IRM penalty is applied (Krueger et al., 2020). More details on the setup can be found in Appendix C.1.

**Experimental Results.** Fig. 2 shows the performance of IRM and IRM with balanced classes in each domain. We make the following observations: first, the test accuracy of IRM drops dramatically under strong $\Lambda$ spuriousness. Second, we show that a naïve fix for IRM, balancing the two classes in each domain by oversampling the minority class, does not lead to improvement in performance.

Table 2: Theoretical validation/test accuracy with the three spurious feature representations (C: color, E: domain and E+C) and the causal feature representation (S: shape) on CMNIST+. The ones with the best validation accuracy in each setting are highlighted, which show the type of feature representations would be learned by ERM and IRM.

| | Validation/Test Accuracy | | | | | | | |
|---|---|---|---|---|---|---|---|---|
| | without label balancing | | | | with label balancing | | | |
| $\rho$ | $\hat{P}(Y\|C)$ | $\hat{P}(Y\|\hat{E})$ | $\hat{P}(Y\|\hat{E},C)$ | $\hat{P}(Y\|S)$ | $\hat{P}(Y\|C)$ | $\hat{P}(Y\|\hat{E})$ | $\hat{P}(Y\|\hat{E},C)$ | $\hat{P}(Y\|S)$ |
| 0.55 | 0.662/0.2 | 0.646/0.35 | 0.662/0.35 | **0.75/0.75** | 0.662/0.2 | 0.5/0.5 | 0.662/0.2 | **0.75/0.75** |
| 0.6 | 0.7/0.2 | 0.68/0.35 | 0.7/0.35 | **0.75/0.75** | 0.7/0.2 | 0.5/0.5 | 0.7/0.2 | **0.75/0.75** |
| 0.65 | 0.738/0.2 | 0.714/0.35 | 0.738/0.35 | **0.75/0.75** | 0.738/0.2 | 0.5/0.5 | 0.737/0.2 | **0.75/0.75** |
| 0.7 | **0.775/0.2** | 0.748/0.35 | **0.775/0.35** | 0.75/0.75 | **0.775/0.2** | 0.5/0.5 | **0.775/0.2** | 0.75/0.75 |
| 0.8 | **0.85/0.2** | 0.815/0.35 | 0.815/0.35 | 0.75/0.75 | **0.85/0.2** | 0.5/0.5 | **0.85/0.2** | 0.75/0.75 |
| 0.85 | **0.888/0.2** | 0.849/0.35 | 0.849/0.2 | 0.75/0.75 | **0.888/0.2** | 0.5/0.5 | **0.888/0.2** | 0.75/0.75 |
| 0.9 | **0.925/0.2** | 0.883/0.35 | 0.883/0.2 | 0.75/0.75 | **0.925/0.2** | 0.5/0.5 | **0.925/0.2** | 0.75/0.75 |

## 3.2 WHY DOES IRM WORK FOR THE ORIGINAL COLORED MNIST?

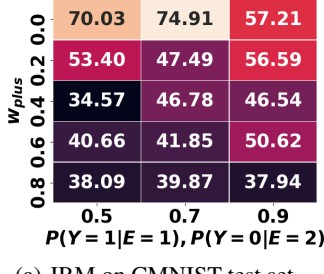

(a) IRM on CMNIST test set

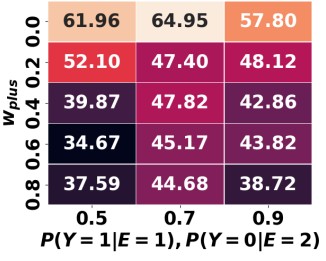

(b) IRM on CMNIST+ test set

Figure 4: Accuracy of IRM trained on datasets between CMNIST and CMNIST+ and tested on the test sets of CMNIST and CMNIST+ ($w_{plus}$ increases): As the training set becomes more similar to CMNIST+, the performance of IRM gradually drops, on the test sets of CMNIST and CMNIST+.

The Colored MNIST (CMNIST) dataset cannot expose the limitation of IRM under strong $\Lambda$ spuriousness. This is because its two training domains are quite similar. As shown in the large triangle on the left in Fig. 3, the values of $P(C|Y,E)$ are similar for $E = 1$ and $E = 2$. In addition, the values of $P(Y|E)$ are the same for all $E$. This makes it difficult to satisfy the IRM constraint by learning feature representations $F(X) \approx E$.

Then, we present experiments to show how IRM gradually goes from working on CMNIST to failing on CMNIST+. In Fig. 3 Left and Right, we observe the differences between between CMNIST and CMNIST+ ($\rho = 0.9$). In Fig. 3 Middle, we create various datasets that interpolate between these two datasets, illustrated by the yellow line in the middle triangle. We use a parameter $w_{plus} \in [0, 1]$

to control $P(C|Y, E)$ of the weights of CMNIST+ ($\rho = 0.9$) in the interpolated dataset:

$$P(C|Y, E) = \frac{P_{\text{cmnist+}}(C|Y, E)w_{plus} + P_{\text{cmnist}}(C|Y, E)(1 - w_{plus})}{\sum_C (P_{\text{cmnist+}}(C|Y, E)w_{plus} + P_{\text{cmnist}}(C|Y, E)(1 - w_{plus}))}, \tag{2}$$

where $P_{\text{cmnist+}}(C|Y, E)$ and $P_{\text{cmnist}}(C|Y, E)$ are the values of $P(C|Y, E)$ in CMNIST+ and CM-NIST. When $w_{plus} = 0$ and $P(Y = 1|E = 1) = P(Y = 0|E = 2) = 0.5$, the dataset is the same with CMNIST. As $w_{plus}$ and $P(Y = 1|E = 1) = P(Y = 0|E = 2)$ increase, the dataset becomes more similar to CMNIST+, and becomes the same with CMNIST+ ($\rho = 0.9$) when $w_{plus} = 1$ and $P(Y = 1|E = 1) = P(Y = 0|E = 2) = 0.9$. For the training sets, we set $P(Y = 1|E = 1) = P(Y = 0|E = 2) \in \{0.5, 0.7, 0.9\}$. Fig. 4 shows results of IRM on the test sets of both CMNIST and CMNIST+. The performance of IRM gradually drops when the values of $P(C|Y, E)$ becomes more similar to CMNIST+ (when the $\Lambda$ spuriousness becomes stronger).

# 4 AN EFFECTIVE FIX FOR IRM

In this section, we propose a simple but effective solution to address the limitation of the IRM training. In a series of experiments, we show that the proposed method can improve the performance of IRM even under strong $\Lambda$ spuriousness.

Since the IRM constraint is too general, we propose an effective solution – combining the conditional distribution matching (CDM) constraint and the IRM constraint. The CDM constraint (Li et al., 2018; Long et al., 2018) requires $P(F(X)|Y, E) = P(F(X)|Y)$, which means the feature representation distribution given the class label should be invariant across different domains. We first explain why this can be a reasonable solution. Then, we propose two types of models that combine the two constraints.

Here, we use the two cases from Section 3.1 to explain why combining IRM with CDM can be an effective fix for IRM.

**Perfectly predicting the label satisfies CDM.** In this case, $F(X) = Y$ and we already know the IRM constraint is satisfied. The CDM condition also holds as $P(F(X)|Y, E) = P(Y|Y, E) = P(Y|Y) = P(F(X)|Y)$.

**Perfectly predicting the domain label violates CDM.** In this case, $F(X) = E$ and the IRM constraint can be satisfied, but the CDM constraint does not hold as $P(F(X)|Y, E) = P(E|Y, E) \neq P(E|Y) = P(F(X)|Y))$. This case implies that we can add CDM to IRM to exclude the undesirable solution of learning spurious features that accurately predict the domain label.

One way to enforce the CDM constraint is through adversarial training: a discriminator tries to infer the source domain from the feature representation $F(X)$ and $F(X)$ tries to adjust itself to fool the discriminator. From this perspective, we can find that the CDM constraint exactly prevents $F(X)$ from making use of the domain labels to achieve high training accuracy (i.e., low $R_e$).

With the idea of combining IRM and CDM, we can formulate the optimization problem as:

$$\underset{\boldsymbol{\theta}_F, \boldsymbol{\theta}_G}{\arg\min} \sum_{e \in \mathcal{E}_{tr}} \mathbb{E}_{(x,y) \sim D_e}[R^e(G(F(x)), y)]$$
$$s.t. \ Y \perp\!\!\!\perp E|F(X), P(F(X)|Y, E = e) = P(F(X)|Y, E = e'), e \neq e', \tag{3}$$

where $R^e$ denotes the loss function in domain $e$ (e.g., cross entropy loss). To impose the CDM constraint, we aim to minimize certain divergence between feature distributions from different domains, denoted as $div(P(F(X)|Y, E = e)||P(F(X)|Y, E = e')), e \neq e'$.

We propose two choices of divergences, i.e., Maximum Mean Discrepancy (MMD) and Kullback-Leibler (KL) divergence (through adversarial training), which result in two algorithms: IRM-MMD and IRM-ACDM (i.e., IRM-Adversarial Conditional Distribution Matching).

**IRM-MMD.** In IRM-MMD, we adopt Maximum Mean Discrepancy (MMD) (Long et al., 2015; Tolstikhin et al., 2017; Shalit et al., 2017) as the distribution divergence. The MMD between $P$ and $Q$, two $d$-dimensional distributions of feature representations, can be defined as:

$$MMD_k(P, Q) = \sup_{f \in \mathcal{H}} |\mathbb{E}_{Z \sim P}[f(Z)] - \mathbb{E}_{Z \sim Q}[f(Z)]|, \tag{4}$$

where $Z \in \mathbb{R}^d$, $f : \mathbb{R}^d \to \mathbb{R}$ maps a feature representation to a real value, $k : \mathbb{R}^d \times \mathbb{R}^d \to \mathbb{R}$ denotes the characteristic kernel of $f$ and $\mathcal{H}$ is the RHKS of $k$. The MMD in Eq. 4 is not directly computable. So, we use the unbiased estimator of MMD (Gretton et al., 2012a) with $N$ samples $z_1^P, ..., z_N^P$ from $P$ and $M$ samples $z_1^Q, ..., z_M^Q$ from $Q$:

$$MMD_k(P,Q) = \frac{1}{N(N-1)} \sum_{i \neq j} k(z_i^P, z_j^P) + \frac{1}{M(M-1)} \sum_{i \neq j} k(z_i^Q, z_j^Q) - \frac{2}{MN} \sum_{i=1}^{N} \sum_{j=1}^{M} k(z_i^P, z_j^Q). \tag{5}$$

With MMD defined, we can define the loss function of IRM-MMD as:

$$\arg\min_{\boldsymbol{\theta}} \sum_{e \in \mathcal{E}_{tr}} \mathcal{L}_{IRM}^e + \beta \sum_{y \in \mathcal{Y}} \sum_{e \in \mathcal{E}_{tr}} \sum_{e' \in \mathcal{E}_{tr} \setminus e} MMD_k(P(F(X)|y,e), P(F(X)|y,e')), \tag{6}$$

where $\mathcal{L}_{IRM}^e = \frac{1}{n_e} \sum_{i=1}^{n_e} R^e(G(F(x_i^e)), y_i^e) + \alpha|| \bigtriangledown_{w|w=1.0} wR^e(G(F(x_i^e)))||^2$ denotes the IRM regularized loss (Arjovsky et al., 2019) for domain $e$, hyperparameters $\alpha$ and $\beta$ control the trade-off between the main loss, the IRM constraint and the CDM constraint.

**IRM-ACDM.** In IRM-ACDM, we make $div(P(F(X)|Y, E = e)||P(F(X)|Y, E = e'))$ be the Kullback-Leibler (KL) divergence and use adversarial learning to estimate it (Tolstikhin et al., 2017; Song et al., 2020). More precisely, we define the loss function of IRM-ACDM as:

$$\arg\min_{\boldsymbol{\theta}} \sum_{e \in \mathcal{E}_{tr}} \mathcal{L}_{IRM}^e + \beta \sum_{y \in \mathcal{Y}} \sum_{e \in \mathcal{E}_{tr}} \gamma_e^y KL(P(F(X)|Y = y, E = e)||P(F(X)|Y = y)), \tag{7}$$

where $\gamma_e^y := P(E = e, Y = y)$. In adversarial learning, a conditional discriminator $D : \mathbb{R}^d \times \mathcal{Y} \to \mathcal{E}_{tr}$ with parameters $\boldsymbol{\theta}_D$ is introduced to predict the domain label of an instance, given its feature representation and class label. As proved in Li et al. (2018); Song et al. (2020), the estimation of the KL divergence above can be reformulated as the minimax game below:

$$\min_{\boldsymbol{\theta}} \max_{\boldsymbol{\theta}_D} \sum_{e \in \mathcal{E}_{tr}} \mathcal{L}_{IRM}^e + \beta \sum_{y \in \mathcal{Y}} \sum_{e \in \mathcal{E}_{tr}} \gamma_e^y \mathbb{E}_{F(x) \sim P(F(X)|Y=y, E=e)}[\log D^e(F(x), y)], \tag{8}$$

where $D^e(F(x), y) = \hat{P}(E = e|F(x), y)$ is the predicted probability of the instance for domain $e$ by the discriminator $D$. In practice, we solve the minimax game above efficiently by the alternative gradient ascent/descent algorithm.

**Experimental Results.** We evaluate IRM-MMD and IRM-ACDM on CMNIST+ to show their efficacy under strong $\Lambda$ spuriousness. We let the output of the second last layer of LeNet-5 to be the feature representation $F(X)$. For IRM-MMD, we use the multiple kernel MMD (MKMMD) (Gretton et al., 2012b; Long et al., 2015). For IRM-ACDM, we use a two-layer MLP with $|\mathcal{E}_{tr}|$ outputs as the discriminator $D$. To show that CDM alone cannot solve OOD prediction under strong $\Lambda$ spuriousness, we set the weight of the IRM penalty $\alpha = 0$ in IRM-MMD and IRM-ACDM to obtain the models regularized by MMD and ACDM, respectively. We consider EIIL (Creager et al., 2020) that solves a minimax game. In the max step, it learns soft domain labels for instances s.t. the IRMv1 loss is maximized. In the min step, it minimizes the IRMv1 loss. We also include ERM and oracle. The oracle uses the original LeNet-5 architecture with a single-channel CNN as the first layer. It is trained and tested with instances transformed into grayscale.

Table. 3 shows the performance of IRM-MMD, IRM-ACDM and the baselines. Under strong $\Lambda$ spuriousness, compared to IRM (Fig. 2), we can observe the significant and consistent performance improvement over IRM resulting from combining CDM with IRM. IRM-MMD and IRM-ACDM also outperform MMD and ACDM. This verifies that CDM alone cannot solve the OOD prediction problem. EIIL reaches comparable accuracy to IRM-MMD and IRM-ACDM when $\rho = 0.85, 0.9$. But EIIL has larger standard deviation $\pm 10.32\%, \pm 13.03\%, \pm 13.03\%$ with $\rho = 0.8, 0.85, 0.9$, compared to IRM-ACDM ($\pm 4.34\%, \pm 4.65\%, \pm 3.09\%$) and IRM-MMD ($\pm 4.56\%, \pm 2.50\%, \pm 6.97\%$). Discussion can be found in Appendix C.3.

## 5 RELATED WORK

**OOD Prediction.** IRM (Arjovsky et al., 2019) formulates causal feature learning as a constraint on the ERM framework (Vapnik, 1992), which imposes the causal inductive bias: causal feature

Table 3: Test accuracy on CMNIST+: by combining CDM with IRM, performance is improved significantly and consistently under strong $\Lambda$ spuriousness ($\rho \geq 0.8$) compared to IRM.

| Method | $\rho = 0.8$ | $\rho = 0.85$ | $\rho = 0.9$ |
|---|---|---|---|
| IRM-MMD (ours) | **52.91%** | **40.83%** | **37.96%** |
| IRM-ACDM (ours) | **57.23%** | **45.47%** | **42.85%** |
| EIIL | 43.40% | 43.24% | 40.93% |
| IRM | 47.01% | 37.81% | 36.31% |
| MMD | 23.04% | 25.22% | 24.22% |
| ACDM | 30.41% | 29.48% | 25.53% |
| ERM | 30.16% | 27.83% | 24.61% |
| Oracle | 73.10% | 73.49% | 73.58% |

representations lead to the existence of an optimal classifier for all domains. Then it is transformed to a regularizer which can be minimized along with empirical risks. Chang et al. (2020) propose a variant of IRM. They implement IRM by minimizing the difference between two classifiers' outputs, which take $F(X)$ and $Concat(F(X), E)$ as the inputs, respectively. Ahuja et al. (2020) reformulate the optimization problem of IRM from a game theory aspect. IRM is related to robust optimization (Ben-Tal et al., 2009). The goal is to optimize the worst domain specific empirical risk. Krueger et al. (2020) extend robust optimization (Ben-Tal et al., 2009) to minimize the empirical risk of the worst domain and maximize that of other domains. They also propose to minimize the variance of domain specific empirical risks along with the empirical risk. Jin et al. (2020) propose to minimize the domain specific risk between two models, one trained on the same domain, the other trained on the other domains. These methods essentially minimize the differences among domain specific risks. Moreover, data augmentation can also improve the generalizability of DNNs from the data perspective (Ilse et al., 2020; van der Wilk et al., 2018). However, it requires prior knowledge on the differences between training and test domains, which are not be available in OOD prediction. Kuang et al. (2018) and Qiao et al. (2020) aim to handle the case with only one training domain. Zhang et al. (2020b) propose to make DNNs more robust against test data generated by unseen interventions. They model interventions in training data with a generative model and perform test-time inference to catch unseen interventions. Compared to the existing work, this work exposes and fixes the issue of IRM in OOD prediction under strong $\Lambda$ spuriousness.

**Domain Adaptation (DA)** assumes that the unlabeled test set can be used during training and validation. From the methodology aspect, distribution matching methods used in DA, such as gradient reversing (Ganin & Lempitsky, 2015) and adversarial CDM (Long et al., 2018) are useful for OOD prediction. Peters et al. (2016) realize the invariance of causal relationships can be used for DA. Zhang et al. (2013) propose reweighting and kernel based distribution matching methods to handle three types of DA problems: target shift, conditional shift and generalized target shift. Gong et al. (2016) work on extracting transferable components $F(X)$ that ensure $P(F(X)|Y)$ to be invariant across domains with location-scale transformation. Their method can identify how $P(Y)$ changes across domains simultaneously. Different from DA, we strictly ensure that the test domain is unseen during training and validation to reflect the scenario of OOD prediction in real-world applications.

## 6 CONCLUDING REMARKS

This work focuses on the OOD prediction problem under strong $\Lambda$ spuriousness. Strong $\Lambda$ spuriousness means the correlations between spurious features and the class label are strong. We find an important limitation of IRM in OOD prediction under strong $\Lambda$ spuriousness: it can be satisfied by spurious feature representations that are predictive of the domain label. To verify it, we design the CMNIST+ dataset which has strong $\Lambda$ spuriousness between color (spurious features) the class label through their common cause – the domain variable. On CMNIST+, we observe the performance of IRM dramatically drops when the $\Lambda$ spuriousness becomes stronger. Based on this observation, we propose a simple but an effective fix to mitigate this issue of IRM. The proposed approach combines CDM and IRM because CDM can also be satisfied by causal feature representations. At the same time, CDM can prevent DNNs from picking up the aforementioned spurious feature representations. Experimental results on CMNIST+ show significant performance improvement of the proposed method, demonstrating its effectiveness. Interesting future work includes (1) extension of the proposed method to OOD prediction tasks in complex data (e.g., graphs and time series) and (2) development of general causal inductive bias that can impose various conditional independence.

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

## A  IRM AND THE CONDITIONAL INDEPENDENCE

Under general conditions, the conditional independence $Y \perp\!\!\!\perp E|F(X)$ is a necessary condition for solutions of the original IRM optimization problem. In (Arjovsky et al., 2019), IRM is defined as a two-stage optimization problem. Any solution $F(X)$ to the IRM optimization problem must satisfy $Y \perp\!\!\!\perp E|F(X)$. However, some $F(X)$ satisfying $Y \perp\!\!\!\perp E|F(X)$ may not be a solution to the original IRM problem. For example, generally, $F(X) = E$ would not minimize the sum of the domain-specific risk $R^e$. However, under strong $\Lambda$ spuriousness, there exist solutions to the original IRM problem that still pick up spurious features. Consider the extreme case, in the training data, if $Y = E$, then $F(X) = E$ is a solution to the problem.

## B  ANALYSIS FOR CMNIST+

*Fitting Color for Classification.* In expectation, the ERM model would learn color as the feature representation for classification. Here, we show the theoretical results for such cases. When the model learns color $C$ as the feature representation $F(X)$, we have:

$$
\begin{aligned}
P(Y|C) &= \sum_E P(Y|C,E)P(E|C) \\
&= \sum_E \frac{P(C|Y,E)P(Y|E)}{P(C|E)}P(E|C) \\
&= \sum_E P(C|Y,E)P(Y|E)\frac{P(E)}{P(C)}
\end{aligned}
\tag{9}
$$

The second equality is by Bayes' rule. We know that $P(C) = \sum_Y \sum_E P(C|Y,E)P(Y|E)P(E)$. Then, given $P(C|Y,E)$, $P(Y|E)$, $P(C)$ and Eq. 9, we can obtain $P(Y|C)$ as shown in Table 4. These results imply that the deterministic classifier $\hat{P}(Y = 1|C = G) = 1$, $\hat{P}(Y = 1|C = B) = 1$, $\hat{P}(Y = 1|C = R) = 0$ for all $\rho \in [0.55, 0.9]$ as shown in Table 4. So, the accuracy of the deterministic classifier $\hat{P}(Y|C)$ on the test set is 0.2. Its accuracy on the training set is $\sum_C P(\hat{Y} = Y|C)P(C)$ where $P(C) = \sum_E \sum_Y P(C|Y,E)P(Y|E)P(E)$.

*Fitting Domain Label for Classification.* Since the spurious correlation between the domain label and the class label is strong, and IRM cannot penalize models fitting the domain label, $P(Y|E)$ can help us understand the expected behavior of the IRM model. Note that the test data is from an unseen domain. So, we analyze the model that first predicts domain by color, then predicts class label by domain. First, we analyze $P(E|C)$ as below:

$$
\begin{aligned}
P(E|C) &= \sum_Y P(E|C,Y)P(Y|C) \\
&= \sum_Y \frac{P(C|Y,E)P(E|Y)P(Y|C)}{P(C|Y)} \\
&= \sum_Y \frac{P(C|Y,E)P(E|Y)P(Y)}{P(C)} \\
&= \sum_Y \frac{P(C|Y,E)P(Y|E)P(E)}{P(C)},
\end{aligned}
\tag{10}
$$

where the second and forth qualities are by Bayes' rule. With Eq. 10, we can list the values of $P(E|C)$ in Table 6 for $\rho \in [0.55, 0.9]$. Thus, we know the deterministic domain prediction results would be $\hat{P}(E = 1|C = G) = 1$, $\hat{P}(E = 1|C = B) = 0$, and $\hat{P}(E = 1|C = R) = 0$. Since $P(Y|E)$ is given in Table 1, we can obtain the deterministic classifier's predictions as:

$$
\hat{P}(Y = 1|C = G) = 1, \ \hat{P}(Y = 1|C = B) = 0, \ \hat{P}(Y = 1|C = R) = 0. \tag{11}
$$

So, the expected test accuracy of the model would be 0.35. Recall that the model first predicts domain label by color and then predict class label by the predicted domain. By doing this, it would have $F(X) \approx E$. This makes it approximately satisfy the IRM constraint $Y \perp\!\!\!\perp E|F(X)$ since

$Y \perp\!\!\!\perp E|E$. This implies that the model's performance can be treated as the expected performance of the IRM model in CMNIST+. In terms of the performance of $\hat{P}(Y|\hat{E})$ on the training set, we can get the results using the same prediction rules as in Eq. 11.

One may argue that the IRM model can perform well if we balance the two classes in each domain. Here, we theoretically show this is not the case. By setting $P(Y|E) = 0.5$, we can obtain the values of $P(E|C)$ in Table 7. Note that practically this can be done by oversampling the minority class of each domain in each mini-batch. However, since $P(Y|E) = 0.5$, the predictions made by the deterministic classifier $\hat{P}(Y|E)$ would be just random guess, leading to a test accuracy of $0.5$.

*Fitting both Domain and Color for Classification.* Here, we consider the model that first predicts the domain label by color and then predicts the class label by both the color and the predicted domain label. The first step is the same as the model fitting the domain label. For the second step, we analyze $P(Y|C, E)$ as below:

$$P(Y|C, E) = \frac{P(C|Y, E)P(Y|E)}{P(C|E)}. \tag{12}$$

With $P(C|E) = \sum_Y P(C|Y, E)P(Y|E)$ and Eq. 12, we obtain values of $P(Y|C, E)$ as shown in Table 8. So, given the predicted domains, $\hat{P}(E = 1|C = G) = 1$, $\hat{P}(E = 1|C = B) = 0$, and $\hat{P}(E = 1|C = R) = 0$, the predictions on the class label are $\hat{P}(Y = 1|C = G, E = 1) = 1$, $\hat{P}(Y = 1|C = B, E = 2) = 1$, $\rho > 0.8$, $\hat{P}(Y = 1|C = B, E = 2) = 0$, $\rho \leq 0.8$ and $\hat{P}(Y = 1|C = R, E = 2) = 0$. So, the test accuracy is $0.35$ when $\rho \leq 0.8$ and $0.2$ when $\rho > 0.8$.

Similarly, when we make $P(Y|E) = 0.5$ by oversampling the minority class in each domain, we can obtain the predictions shown in Table 9. So, the deterministic model would make predictions as $\hat{P}(Y = 1|C = G, E = 1) = 1$, $\hat{P}(Y = 1|C = B, E = 2) = 1$, and $\hat{P}(Y = 1|C = R, E) = 0$. This would lead to a test accuracy of $0.2$. These results reflect the reasons why the IRM model trained with the balanced classes in each domain ($P(Y|E) = 0.5$) has worse performance compared to its counterpart trained with the original CMNIST+ data.

Table 4: Analysis results: fitting color

| $\rho$ | $P(Y = 1|C = G)$ | $P(Y = 1|C = B)$ | $P(Y = 1|C = R)$ |
|---|---|---|---|
| 0.55 | 0.697 | 0.534 | 0.29 |
| 0.6 | 0.737 | 0.545 | 0.25 |
| 0.65 | 0.775 | 0.56 | 0.212 |
| 0.7 | 0.811 | 0.577 | 0.176 |
| 0.8 | 0.88 | 0.63 | 0.111 |
| 0.85 | 0.912 | 0.67 | 0.081 |
| 0.9 | 0.942 | 0.73 | 0.053 |

Table 5: Analysis results: fitting color, when $P(Y|E) = 0.5$.

| $\rho$ | $P(Y = 1|C = G)$ | $P(Y = 1|C = B)$ | $P(Y = 1|C = R)$ |
|---|---|---|---|
| 0.55 | 0.633 | 0.633 | 0.29 |
| 0.6 | 0.667 | 0.667 | 0.25 |
| 0.65 | 0.702 | 0.702 | 0.212 |
| 0.7 | 0.739 | 0.739 | 0.176 |
| 0.8 | 0.818 | 0.818 | 0.111 |
| 0.85 | 0.86 | 0.86 | 0.081 |
| 0.9 | 0.905 | 0.905 | 0.053 |

## C  EXPERIMENTAL SETUP AND RESULTS

Here, we include more details and discussion on experimental setup, datasets and results.

Table 6: Theoretical analysis results: predicting domain by color.

| $\rho$ | $P(E=1|C=G)$ | $P(E=1|C=B)$ | $P(E=1|C=R)$ |
|---|---|---|---|
| 0.55 | 0.697 | 0.466 | 0.332 |
| 0.6 | 0.737 | 0.455 | 0.3 |
| 0.65 | 0.775 | 0.44 | 0.27 |
| 0.7 | 0.811 | 0.423 | 0.241 |
| 0.8 | 0.88 | 0.37 | 0.189 |
| 0.85 | 0.912 | 0.33 | 0.165 |
| 0.9 | 0.942 | 0.27 | 0.142 |

Table 7: Theoretical analysis results: predicting domain by color, when $P(Y|E)=0.5$.

| $\rho$ | $P(E=1|C=G)$ | $P(E=1|C=B)$ | $P(E=1|C=R)$ |
|---|---|---|---|
| 0.55 | 0.633 | 0.367 | 0.5 |
| 0.6 | 0.667 | 0.333 | 0.5 |
| 0.65 | 0.702 | 0.298 | 0.5 |
| 0.7 | 0.739 | 0.261 | 0.5 |
| 0.8 | 0.818 | 0.182 | 0.5 |
| 0.85 | 0.86 | 0.14 | 0.5 |
| 0.9 | 0.905 | 0.095 | 0.5 |

Table 8: Predicting by both color and domain

| $\rho$ | $P(Y=1|G,1)$ | $P(Y=1|G,2)$ | $P(Y=1|B,1)$ | $P(Y=1|B,2)$ | $P(Y=1|R,1)$ | $P(Y=1|R,2)$ |
|---|---|---|---|---|---|---|
| 0.55 | 0.957 | 0.1 | 0.9 | 0.214 | 0.786 | 0.043 |
| 0.6 | 0.964 | 0.1 | 0.9 | 0.25 | 0.75 | 0.036 |
| 0.65 | 0.971 | 0.1 | 0.9 | 0.292 | 0.708 | 0.029 |
| 0.7 | 0.977 | 0.1 | 0.9 | 0.341 | 0.659 | 0.023 |
| 0.8 | 0.986 | 0.1 | 0.9 | 0.471 | 0.529 | 0.014 |
| 0.85 | 0.99 | 0.1 | 0.9 | 0.557 | 0.443 | 0.01 |
| 0.9 | 0.994 | 0.1 | 0.9 | 0.667 | 0.333 | 0.006 |

Table 9: Predicting by both color and domain, class balanced

| $\rho$ | $P(Y=1|G,1)$ | $P(Y=1|G,2)$ | $P(Y=1|B,1)$ | $P(Y=1|B,2)$ | $P(Y=1|R,1)$ | $P(Y=1|R,2)$ |
|---|---|---|---|---|---|---|
| 0.55 | 0.71 | 0.5 | 0.5 | 0.71 | 0.29 | 0.29 |
| 0.6 | 0.75 | 0.5 | 0.5 | 0.75 | 0.25 | 0.25 |
| 0.65 | 0.788 | 0.5 | 0.5 | 0.788 | 0.212 | 0.212 |
| 0.7 | 0.824 | 0.5 | 0.5 | 0.824 | 0.176 | 0.176 |
| 0.8 | 0.889 | 0.5 | 0.5 | 0.889 | 0.111 | 0.111 |
| 0.85 | 0.919 | 0.5 | 0.5 | 0.919 | 0.081 | 0.081 |
| 0.9 | 0.947 | 0.5 | 0.5 | 0.947 | 0.053 | 0.053 |

## C.1 EXPERIMENTAL SETUP

Here, we provide more details on experimental setup. We perform grid search for hyperparameter tuning. For IRM, IRM-MMD and IRM-ACDM, we search the iteration number to plug in the IRM penalty term ($K_{IRM}$) in $200, 400, 600$ and IRM penalty weight $\alpha$ in $\{1, 10, ..., 10^8\}$. For MMD, ACDM, IRM-MMD, IRM-ACDM, we search CDM penalty weight $\beta$ in $\{1, 10, ..., 10^5\}$. For ACDM and IRM-ACDM, we set the number of steps we train the discriminator $D$ in each iteration to 10. For EIIL, we do the same hyperparameter tuning on the IRM penalty weight ($\alpha$), the iteration to add IRM ($K_{IRM}$) as we do for IRM, IRM-ACDM and IRM-MMD methods. We train the soft environment weight $q(E|X,Y)$ for $\{10, 100, 1000, 10000\}$ steps.

## C.2 DATASETS

Here, we present more details about the datasets.

**CMNIST.** CMNIST is introduced by (Arjovsky et al., 2019). We can see the two training domains of CMNIST are similar to each other in terms of both $P(C|Y,E)$ and $P(Y|E)$ in Table 10. This means CMNIST does not cover the case of strong $\Lambda$ spurious, since the spurious correlations, color–domain and domain–class, are not strong.

Table 10: Description of CMNIST

| $E$ | $P(Y=1\|E)$ | $Y$ | $P(C=G\|Y,E)$ | $P(C=B\|Y,E)$ | $P(C=R\|Y,E)$ |
|---|---|---|---|---|---|
| $E=1$ | 0.5 | $Y=1$ | 0.9 | 0.0 | 0.1 |
| | | $Y=0$ | 0.1 | 0.0 | 0.9 |
| $E=2$ | 0.5 | $Y=1$ | 0.8 | 0.0 | 0.2 |
| | | $Y=0$ | 0.2 | 0.0 | 0.8 |
| $E=3$ | 0.5 | $Y=1$ | 0.1 | 0.0 | 0.9 |
| | | $Y=0$ | 0.9 | 0.0 | 0.1 |

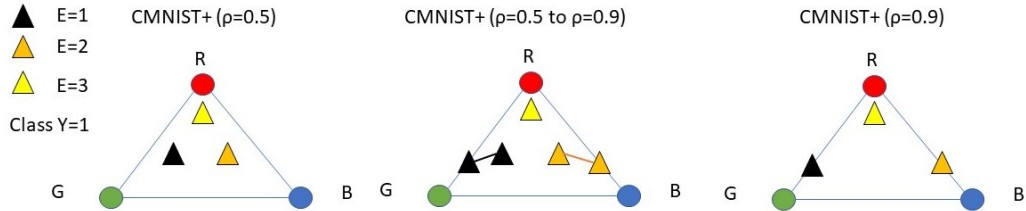

Figure 5: We visualize CMNIST+ with $\rho \in [0.5, 0.9]$ in terms of $P(C|Y=1, E)$. Each large triangle represents the space of $P(C|Y,E)$. Each small triangle shows the values of $P(C=c|Y=1, E=e), c \in \{R, G, B\}, e \in \{1, 2, 3\}$.

**CMNIST+.** We visualize the CMNIST+ dataset with different values of $\rho$ in Fig. 5. In addition, we provide a detailed simulation recipe of CMNIST+ and compare it with that of CMNIST. This would also show that CMNIST+ is in accordance with the causal graph in Fig. 1.

1. We decide the true label $Y^*$ (without noise) of each instance by its original digit label (0-9).

2. We randomly split the data into test and training.

3. We assign the training instances to the two training domains based on the true label $Y^*$ and $P(Y|E)$ in Table 1. This step introduces correlations between $Y$ and $E$. In each training environment, we further randomly split the data into training and validation.

4. In each environment, we generate noisy labels $Y$ by randomly flipping them with 25% probability. This means $Y^* \to Y$.

5. Given $P(C|Y,E)$ in Table 1, the noisy label and the domain label, we assign color to each instance. This step introduces correlations among $X^s$, $Y$ and $E$.

Note that there is a difference in what causal relationships mean in traditional causal inference and in OOD prediction. In OOD prediction, the definition of causal relationships is different from a traditional one. Traditionally, $X^c \to Y$ means the generation of $Y$ is (partially) determined by $X^c$. It does not necessarily mean $P(Y|X^c)$ remains the same across domains (Pearl & Bareinboim, 2014). However, in OOD prediction, we say there exists a causal relationship $X^c \to Y$ iff $P(Y|X^c)$ is the same across different domains. We also know in the original MNIST dataset, there exists invariant causal relationship $X^c \to Y^*$. This implies that, from the data generating process of CMNIST+, we confirm that (1) there exist causal relationships $X^c \to Y^* \to Y$, (2) there are correlations among $X^s$, $Y$ and $E$. With these two conclusions, we can claim that the causal graph in Fig. 1 is in accordance with CMNIST+ in the OOD prediction problem where the true label $Y^*$ is ignored as it is not used in training and evaluation.

**Creating Strong $\Lambda$ Spuriousness with Two Colors.** It is possible to setup strong $\Lambda$ spuriousness with two colors for binary classification with two training domains. Here, we use the two colors: red (R) and green (G). To show it is possible, we use an example with the following setup: $P(Y=1|E=1) = 0.9$, $P(Y=1|E=2) = 0.1$. Let's say for $E=1$, we set $P(C=G|Y=1, E=1) = 0.9$, $P(C=G|Y=0, E=1) = 0.1$. Then, we can set $P(C=G|Y=1, E=2) = 0.1$, $P(C=G|Y=0, E=2) = 0.9$ for $E=2$. This setup makes strong correlations among the color, the class label and the domain label. Thus, it is possible to create strong $\Lambda$ spuriousness with just two colors. Our concern with such datasets is that even if strong $\Lambda$ spuriousness exists in training

domains, it is a challenge to create test domains that are diverse enough from the training ones. Following the aforementioned setup, for the test domain $E = 3$, if we set $P(Y = 1|E = 3) = 0.5$, $P(C = G|Y = 1, E = 3) = 0.5$ and $P(C = G|Y = 0, E = 3) = 0.5$, it would be right in the middle of the two training domains. Unfortunately in this setting, even if IRM fails, it can be difficult to observe it with the test accuracy. This is because a model perfectly fits the color features can reach $0.5$ test accuracy. This leads to smaller differences between models fitting causal features and those with spurious features. Thus, it becomes more challenging to judge whether a model fails in practice, which explains why we use three colors in CMNIST+.

### C.3 MORE DISCUSSION ON RESULTS OF EIIL

We evaluate EIIL[1] on CMNIST+. EIIL gets test accuracy $43.40 \pm 10.32\%$, $43.24 \pm 13.03\%$, $40.93 \pm 13.03\%$ on CMNIST+ with $\rho = 0.8, 0.85, 0.9$. Compared to IRM-ACDM $57.23 \pm 4.34\%$, $44.83 \pm 4.65\%$, $42.85 \pm 3.09\%$ and IRM-MMD $52.91 \pm 4.56\%$, $40.83 \pm 2.50\%$, $37.96 \pm 6.97\%$, EIIL has comparable mean accuracies when $\rho = 0.85, 0.9$, but is not stable. There are two reasons. (1) EIIL relies on IRM and the soft domain weight $q(E|X, Y)$. When $q(E|X, Y)$ takes values s.t. strong $\Lambda$ spuriousness exists, it makes IRM fail. (2) It has issues in model selection. EIIL can only select models by validation accuracy as $q(E|X, Y)$ is a scalar $q_i$ for instance $i$. It is learned for the val/test, so validation loss is not feasible. Validation accuracy can be high when the model picks up spurious features. While model selection with validation loss considers the regularizers (IRM and CDM), which approximates how well models fit causal features.

---

[1]https://github.com/ecreager/eiil

