# OpenReview forum: "Out-of-distribution Prediction with Invariant Risk Minimization: The Limitation and An Effective Fix"
_ICLR.cc/2021/Conference — Reject_

### Official Review · AnonReviewer3 · 2020-10-27
**Paper proposes a new dataset extending CMNIST to highlight flaws of IRM  and proposes a fix. Existing works that extend IRM already suffice to fix this problem. Several theoretical issues with fix proposed by the authors.**

**Rating:** 4
**Confidence:** 5

**Review:**

Summary:
In this work, the authors focus on the out-of-distribution generalization problem. The input is dataset from multiple environments and the goal is to learn a model that generalizes well to an unseen test environment. The work is based on recent line of works on invariant risk minimization (IRM) (Arjovsky et al.). The authors show that under a certain type of structure for the generative model, where the domain/environment label itself has a strong correlation with the spurious factors and the target label, IRM fails. The authors propose an extension of the colored MNIST dataset to highlight this problem. Finally, the authors build a method that works better than IRM on the extension of colored MNIST dataset.

Pros:
 I appreciate the authors have tried to highlight how the IRM directly applied to datasets from multiple environments will not always work and one has to be careful about the environment induced correlations themselves.

Cons:
I divide my concerns into different subsections below.

a)	Incorrect connection between IRM (Arjovsky et al.) and conditional independence made by the authors:

Conditional independence (Y \perp E | F(X)) is a necessary condition but not sufficient for the theory of IRM to work.  Therefore, analyzing any F that satisfies this property is not sufficient.
Suppose we have two training environments, E=1 and E=2.  Asssume that we are only interested in binary classification for now. The main condition that is assumed in Arjovsky et al. (Page 9 Definition 7) for the success of IRM is that there exists a representation F*(X) such that

P(Y|F*(X),E=0) = P(Y|F*(X), E=1) = P(Y|F*(X))

The above condition implies that Y \perp E | F*(X). In the above expression equating conditionals, there is already an assumption made about F*(X), which is that for both environments E=0 and E=1, the support of F*(X)|E=0 and support of F*(X)|E=1 are equal. Suppose the supports were not equal, then the conditionals can only be equated over the intersection of the supports.
In the driving example used in the paper, i.e., F*(X)=E, the support of the two conditionals F*(X)|E=0 -->E=0 and F*(X)|E=1-->E=1 do not intersect.  Therefore, what authors claim is a problem with IRM is not really a problem but a data generating environment for which the theory of IRM is not guaranteed to be successful. The right claim to make is that CMNIST+ does not satisfy the assumptions IRM makes for the method to be successful. However, this is easy to fix as I explain soon.

Before moving to the next section, I would like to also make another important remark. The authors investigate any predictor that satisfies the conditional independence condition. This is also not correct because IRM and other IRM based methods select one of the invariant predictors and not all (thus there can always be bad invariant predictors, which does not mean that they will be selected). For a complete characterization of invariant predictors in terms of conditional independences please refer to Koyama et al.

Update post discussions: This point a) was corrected by the authors.

b)   Why does IRM not work on the CMNIST+?

We now turn to providing the explanation why IRM did not work on CMNIST+ as the explanation provided by the authors is not accurate. As we explained in the last section, F(X)=E is a representation that does not satisfy the criterion that the theory of IRM requires. One intuitive way to think is that the success of IRM assumes that representation F that we search over have an overlapping support across the environments. The IRM optimization procedure fails because it does not enforce this assumption in any way and F(X)=E can lead to a better predictor than F(X)=S, where S is the true causal feature.
In CMNIST+, the authors have created two environments, where the environment label itself is strongly correlated with the label. Say in E=0, the majority of the labels are 0, and in E=1 the majority of the labels are 1.  Suppose the IRM optimization (Arjovsky et al.) is given a representation F(X)=E as input. The support of F(X) gets partitioned into two disjoint sets X0 = F^{-1}(0) and X1 = F^{-1}(1). E=0 learns a predictor over the set X0 and E=1 learns a predictor over the set X1. A predictor that labels all points in X0 as 0 and X1 as 1 actually satisfies the definition of invariant predictor because it simultaneously minimizes the error in the two environments. If the error of this predictor is actually less than the error of the predictor based on causal features, then this predictor can be selected by the IRM optimization, which is exactly the case in the CMNIST+ dataset.

The main reason IRM was designed was to make the predictors from different environments be compared when the sets X0 and X1 overlap to some extent at least, i.e. the image of F over the feature distributions in the two environments has to overlap.
If the image of F over feature distribution does not overlap at all (as is the case in example considered in the paper), a trivial invariant predictor which is not robust to distribution shifts will exist.


c)	There is a simple alternate fix for the entire problem:
The space of problems that authors want to fix abstractly stated are when the environment label (domain label) itself is so strongly correlated with the label that the IRM is encouraged to use environment as a representation.
The fix goes as follows:  Mix the data from the two environments. Take the mixed data and divide into two completely new environments. A manual way to construct these new environments is to divide the data in such a way that the proportion of the colors in the two environments marginally different as was the case in colored MNIST. For an algorithmic approach  to construct these new environments  use this approach (paper http://www.gatsby.ucl.ac.uk/~balaji/udl2020/accepted-papers/UDL2020-paper-045.pdf) (code available at https://github.com/ecreager/eiil). The above paper shows that a single dataset can be divided into multiple environments and retains the gains shown in IRM.
The two new environments obtained from the algorithm in the above paper or even through a manual division as explained above would lead to a dataset that is very similar to the original CMNIST. Observe that by mixing and creating two new environments, we are automatically ensuring that the support overlap assumptions required by IRM are satisfied.
I believe that these approaches can bring the performance back to 68 percent level.

Also, note that the fix I am proposing is not the simple fix based on label balancing that the authors show does not work. By mixing the environments, we are destroying the spurious environment based correlation that exist.

Therefore, whenever there is a strong environment based spurious correlation, i.e., each environment has a stark difference in the marginal distribution of labels, then the prudent thing to do is to mix the environments destroy the spurious correlation and then construct environments for IRM either manually or through the algorithm I shared above.

Update post discussions: The authors did more experiments to show their method works on CMNIST+ better than these baselines. However, I have major concerns with the principle proposed by the authors as a search criterion. I believe the authors approach happens to work on CMNIST+ but is not based on the right principle. More on this in my point d) below.

d)	The fix proposed by authors has theoretical problems:
The authors have proposed to do conditional distribution matching which constraints that the representation learned has to be independent of the environment conditional on the label. For the sake of discussion, let us consider the structural equation model (SEM) used in Theorem 9 in Arjovsky et al. If we assume that the lambda-spuriousness condition holds for the SEM under consideration. Say for the two environments, the support of the feature distributions do not intersect, the support of the label distributions do not intersect. In this case of lambda-spuriousness, IRM continues to be able to recover the ideal invariant predictor and does not fail (as no assumption in Theorem 9 is violated). However, the representation used by the ideal predictor does not have to satisfy the CDM condition stated by the authors. Therefore, the CDM condition proposed by the authors holds in the extreme case of the example discussed by the authors but does not hold in general.

In other words, author should show how imposing CDM works in a broad range of settings and not just one example that they use. Based on my argument I above, I highly doubt that CDM condition is actually a necessary condition for the success of IRM under lambda-spuriousness.

e)	Colored MNIST + issues:
The authors never explained in detail how the data in the different environments is generated. The Table 1 is an incomplete definition. It only lays down the conditional distributions. Conditional distributions are not sufficient to decipher the underlying structural equation model.  What authors call lambda-spuriousness is not really a lambda structure. In CMNIST+, there has to be an arrow from the label Y to the spurious features Xs as well, which does not appear in lambda structure.

Update based on discussion: The authors explained the data generation process.

f)	A simple comparison with balance in generation time:
Why did the authors not compare their method when P(Y=1|E)=0.5 in both environments. It seems for high values of rho and P(Y=1|E)=0.5 a direct application of IRM does not work.  My suspicion is when  P(Y=1|E)=0.5, IRM does not work because we need more than two training environments, as the number of colors that enter the equation of spurious correlations go from two to three.

g) Comparison with Koyama et al.: Koyama et al. had worked with a lambda type structure and introduced an extension of colored MNIST where environment label plays an important role. A comparison with that work would have been useful.

Update based on discussion: The authors clarified that this paper was posted on arxiv on Aug 4 and the ICLR policy requires them to compare only until Aug 2. I am ok with authors not including a comparison with this paper.

Quality: Unfortunately, the paper is not good quality. A lot more work is needed to really justify why what they propose is really a problem with IRM and why the simple fixes I propose won't already solve the problem.

Significance: The area of OoD generalization is quite significant. However, the problem proposed by the authors and the approach taken by the authors is not of much significance.

Originality: The authors have proposed a new CMNIST dataset and a new algorithm to fix it. The authors should get credit for proposing the dataset but besides that I don't think the algorithm proposed as a fix is needed.


**Final update:** The main criterion used by the authors to search invariant predictors is not correct and is in fact not satisfied by the invariant predictors. For this my suggestion to authors is to modify their criterion in a way that it is at least satisfied by the ideal model you want to learn.

References:

Koyama et al.   "Out-of-distribution generalization with maximal invariant predictor." arXiv preprint arXiv:2008.01883 (2020).

---

> ### Author Response · Authors · 2020-11-18
> **We thank the reviewer for the thoughtful suggestions and extensively detailed reviews.**
>
> ### Relationship between IRM and $Y \perp E | F(X)$
>
> Here, we revise the reviewer’s claim to make it clear. We can say under general conditions, the conditional independence $Y \perp E |F(X)$ is a necessary condition for solutions of the original IRM optimization problem. In the original IRM paper [2], we can see IRM is defined as a two-stage optimization problem. We can see, any solution F(X) to the problem of IRM must satisfy $Y \perp E |F(X)$. However, any F(X) satisfying $Y \perp E |F(X)$ may not be a solution to the original IRM problem. For example, generally, $F(X) = E$ would not minimize the sum of the domain-specific risk $R^e$. However, this does not invalidate our work. **Note that under strong $\Lambda$ spuriousness, there exist solutions to the IRM problem that still pick up spurious features. Consider the extreme case, in the training data, if $Y=E$, then $F(X)=E$ is a solution to the problem.**
>
> ### IRM, strong $\Lambda$ spuriousness and overlapping
>
> IRM [2] never assumes overlapping w.r.t. F(X) across different domains. Instead, the last sentence in the paragraph beneath Definition 3 (Page 5) shows that IRM cannot handle the case $F(X)=E$. This is because IRM can only guarantee to learn feature representations that elicit invariant predictors when there is overlapping in F(X). In short, IRM works under a possibly untenable assumption. Our work points this problem out and proposes a simple and effective fix to it. The meaning of our work is clear.
>
> ### The “simple alternative fix” and the ELLI algorithm from [3]
>
> The simple alternative fix in (c) is based on an incorrect assumption. The reviewer misunderstood the problem setting of OOD prediction. Here, the knowledge of which features are spurious/causal is not given. So, we cannot test whether there exists strong $\Lambda$ spuriousness given the observational data, which should not be taken as prior knowledge. However, this recommended fix -- mixing and redividing the data as proposed must rely on the assumption that we, when constructing our datasets and designing the algorithms, have the prior knowledge that (1) there exists strong $\Lambda$ spuriousness and the (2) color is a spurious correlation.
>
> **The reviewer can propose any method that may or may not work. However, this kind of review is out-of-scope. Even if the reviewer proposes a valid method, it does not invalidate our method. In addition, no evidence shows the simple alternative fix works.**
>
> The workshop paper [3] does not show the method ELLI can solve the OOD prediction problem under strong $\Lambda$ spuriousness. ELLI solves a minimax game. In the max step, it learns to assign a new environment label to each instance s.t. The IRMv1 loss is maximized. In the min step, it minimizes the IRMv1 loss. **Note that in the min step, it still relies on the IRMv1 loss to learn causal features. Once ${q}$ assigns the environments s.t. Strong $\Lambda$ spuriousness exists, either by a bad initialization or a bad local minimum, then ELLI is expected to fail. ELLI relies on a trained reference model that fits specific spurious features, which our method and IRM do not require.**
>
> The reviewer claims hypothetical results without evidence. We believe this is neither a professional nor a scientific way to review papers.
>
> ### The data generating process
>
> In fact, Table 1 is a complete definition of how the data is generated. If not, we hope the reviewer can specify which distribution is not defined.  The data generating process of CMNIST+ is very clear given Table 1 and the value of $\rho$. We closely follow that of the original IRM paper except we modified the values the conditional distributions P(Y|E) and P(C|Y,E) take.
>
> ### Can IRM work when the number of colors is larger than the number of environments?
>
> **Results in Figure 4 ($w_{plus}$ = 0.2) show that IRM can lead to results (~56%) that are significantly better than random guesses when the number of colors is larger than the number of environments and the $\Lambda$ spuriousness is weak. Without defining P(Y|E) and P(C|Y,E) for each environment, the relationship between the number of colors and the number of domains/environments can not be determined.** For example, adding another environment which cannot weaken the strong $\Lambda$ spuriousness is not expected to improve the performance of IRM.
>
> ### Cite and compare with [1]
>
> **Can the reviewer read the reviewer guideline? It says authors are not expected to cite work published after 08/02. [1] is on arxiv on 08/04.**
>
> [1] Koyama, Masanori, and Shoichiro Yamaguchi. "Out-of-distribution generalization with maximal invariant predictor." arXiv preprint arXiv:2008.01883 (2020).
>
> [2] Arjovsky, Martin, Léon Bottou, Ishaan Gulrajani, and David Lopez-Paz. "Invariant risk minimization." arXiv preprint arXiv:1907.02893 (2019).
>
> [3] Creager, Elliot, Jörn-Henrik Jacobsen, and Richard Zemel. "Environment Inference for Invariant Learning." In ICML Workshop on Uncertainty and Robustness. 2020.

---

> > ### Comment · AnonReviewer3 · 2020-11-19
> > **Response**
> >
> > Hi authors
> >
> > You say "The simple alternative fix in (c) is based on an incorrect assumption. The reviewer misunderstood the problem setting of OOD prediction. Here, the knowledge of which features are spurious/causal is not given..." You also say "ELLI relies on a trained reference model that fits specific spurious features, which our method and IRM do not require."
> >
> > a) **Running EIIL (it is not ELLI) does not require knowledge of spurious feature (color for instance), it does not require knowing Lambda spuriousness exists. The reference model q in EIIL  simply assigns the environment and it does not use any "specific spurious features" as you claim so.  All I am asking you to do is run experiments with EIIL  and show that it would not work in the examples you pointed out.  It requires no extra assumption**
> >
> >
> > You say "**The reviewer can propose any method that may or may not work. However, this kind of review is out-of-scope. Even if the reviewer proposes a valid method, it does not invalidate our method. In addition, no evidence shows the simple alternative fix works.** " You also say "The reviewer claims hypothetical results without evidence. We believe this is neither a professional nor a scientific way to review papers."
> >
> > b) **My only goal throughout this process is to provide you with constructive feedback and I hope you take it in good spirit. I also hope you can enforce a good conduct in these conversations too. The most important point here to realize is your paper is based on empirical evidence. I do not think any reviewer provides new experiments as evidence to support their logic.  I had provided a good amount of justification why EIIL (an already existing work) would work, which I restate below.**
> >
> > "Observe that by mixing and creating two new environments, we are automatically ensuring that the support overlap assumptions required by IRM are satisfied. I believe that these approaches can bring the performance back to 68 percent level.
> > Also, note that the fix I am proposing is not the simple fix based on label balancing that the authors show does not work. By mixing the environments, we are destroying the spurious environment based correlation that exist.
> > Therefore, whenever there is a strong environment based spurious correlation, i.e., each environment has a stark difference in the marginal distribution of labels, then the prudent thing to do is to mix the environments destroy the spurious correlation and then construct environments for IRM either manually or through the algorithm I shared above."
> >
> >
> > c) It is easy to check in your examples if Lambda spuriousness exists or not. You simply need to check if the environments and labels bear a high correlation. Besides, as I already said above running EIIL does not require you to make any such assumption.
> >
> > d) About data generation process. In original CMNIST, you follow the following steps.
> > i) Assign each point in MNIST to one of the environments uniformly at random,
> > ii)) Flip the label to add noise,
> > iii) Generate the colors based on environment dependent probability.
> >
> > I would like to see a similar step wise generation process for your data too. In your case, from Table 1 it is not clear the order in which the random variables are generated. This sequence of what is sampled first is important to defining the data generation and also to understanding what are the causal variables (refer to structural equation models in causality to understand why).
> >
> > d) In my point d) of original response I had also raised important theoretical issues regarding your approach and that concern remains.

---

> > > ### Author Response · Authors · 2020-11-23
> > > **Response to theoretical issues in (d)**
> > >
> > > We thank the reviewer for detailed and actionable comments.
> > >
> > > Theorem 9 in [1] crucially relies on the linearity assumption of the modeling. It cannot generalize to the deep learning cases that our paper and many recent papers on IRM focus on. Our main points are: (1) IRM suffers from the non-overlapping issue of $F(X)$ from different domains under strong $\Lambda$ spuriousness with deep models, and (2) IRM-CDM is a simple and effective fix to it. With the linear model assumption, Theorem 9 in [1] does not need overlapping $F(X)$ supports with different $E$. Linear models are global models in which constraints enforced in a local domain are effective globally on the entire space. So, each new domain lying in a linear general position, **although its raw features/feature representations/labels may not overlap with those from other domains**, removes one degree of freedom in the space of invariant solutions **globally for all the other domains**. But for DNNs with universal approximation property, as long as the data from two domains do not overlap, the model has enough capacity to learn a feature extractor s.t. $F(X)$ from different domains do not overlap, making IRM ineffective. This explains IRM’s failure with deep models when different domains have very different $P(F(X)|E)$. In our fix, we enforce the overlapping of $F(X)$ by adding the CDM constraint, its efficacy is shown by results on CMNIST+.
> > > Here, we motivate CDM. For deep models, IRM constraint assumes/requires that $F(X)|E=0$ and $F(X)|E=1$ overlap. This is enforced via CDM which minimizes a certain divergence between $P(F(X)|E=0)$ and $P(F(X)|E=1)$. For example, a finite Jeffreys divergence between two distributions implies that their supports exactly overlap. Specifically, we add the CDM (not the unconditional one) constraint as the mismatch of $F(X)$ from different classes is not desired. Finally, we reformulate the CDM constraint as a regularizer in the optimization as: $\underset{\theta}{\min}   \underset{\theta_D}{\max} \sum_{e}
> > >  \mathcal{L}^e_{IRM} + \beta \mathcal{L}_{CMD}$.
> > > Therefore, the CDM regularizer is a simple and effective fix for the non-overlapping issue of IRM with deep models. From our perspective, it is conceptually simpler than EIIL.
> > >
> > > [1] Arjovsky et al. "Invariant risk minimization." arXiv preprint arXiv:1907.02893 (2019).

---

> > > > ### Comment · AnonReviewer3 · 2020-11-24
> > > > **Response**
> > > >
> > > > Hi
> > > >
> > > > Thanks for the response.
> > > >
> > > > 1. **Concern with conditional independence criterion:**  In my response, the example I gave from Theorem 9 (linear models) was only done for sake of discussion, as I had stated. The point I am making is more general and does not only apply only to linear models. Consider a simple non-linear equation $Y_e \leftarrow f(X_e) + n, X_e$, $Y_e$ are feature, label pair in environment $e$. The ideal model you want to learn is $f(\cdot)$.  The additional conditional independence criterion proposed by the authors as is not satisfied in general by $f(\cdot)$ (there is no reason to believe that $f(X) \perp E |Y$ since Y and X can have arbitrary dependence and only f is invariant across environments).  Suppose $X_1$ and $X_2$ in the two environments come from different parts of the feature space and say also correpsonding images $f(X_1)$ and $f(X_2)$ are also disjoint. In such a case, the label can contain information about both environment and  $f(\cdot)$. In such cases the criterion proposed would not be satisfied.
> > > >
> > > > 2. It is good to see the experiments from EIIL. I believe it is a fairer benchmark.  It seems that the good gains proposed seem to be in a certain range of rho, which is hard to understand.
> > > >
> > > > 3. If $P(Y|X_c)$ is invariant across all the environments, then $X_c$ form the causes of $Y$ (see [1]). Understanding the causal structure of generation of data is useful. It is also standard in CMNIST experiments in the literature on OOD.
> > > >
> > > > My final take on the paper after considering the new experiments and new responses is that the proposed approach is a heuristic. The main criterion and theory motivating the approach itself seem wrong. The approach does seem to perform better than IRM and EIIL. The current form of the paper needs significant work. I would highly recommend authors to think of modifying their criterion into a criterion that is at least always satisfied by the ideal solution.
> > > >
> > > > [1] Bareinboim, E., Brito, C., & Pearl, J. (2012). Local characterizations of causal Bayesian networks. In Graph Structures for Knowledge Representation and Reasoning (pp. 1-17). Springer, Berlin, Heidelberg.

---

> > > > > ### Author Response · Authors · 2020-11-24
> > > > > **We thank the reviewer's response but disagree with the three claims from the reviewer.**
> > > > >
> > > > > ### Response to 1.
> > > > >
> > > > > We strongly disagree with the reviewer’s claim “The main criterion and theory motivating the approach itself **seem wrong**.” In the previous reply, we provide concrete reasons for why Theorem 9 in IRM [1] and the linear model proposed by the reviewer do not invalidate our proposed method. We hope the reviewer can read our previous response carefully.
> > > > >
> > > > > We hope that the reviewer could have a good understanding of the difference between the hard constraint $F(x) \perp E | Y$ and the soft CDM regularization. As mentioned in our paper, the regularization weight $\beta$ controls how much the distributions should be close to each other (overlapping), instead of being strictly equal. The regularization weight $\beta$ should be tuned case by case because how much the hidden feature distributions should overlap differs case by case.
> > > > >
> > > > > In the case $Y_e = f(X_e) + n$, the reviewer mentioned that ideally one wants to learn $f$ even though $X_e$ and $X_{e’}$ do not overlap. First of all, in this case, with neural network modeling $f$, IRM will just learn $f$ independently in each environment. Such independent $f$ functions are not the desired invariant predictor. Second, we hope that the reviewer can understand the meaning of **invariant feature learning**. In this problem, we want to learn deep feature representations $F(X)$, that can bring data from different environments ($X_e$ and $X_{e’}$) into the same hidden space. We make use of the prior information **there is some invariance across domains** by encouraging their features overlapping and enforcing that they share one common predictor (e.g., a linear layer) on the overlap. If their features do not overlap, then to learn a universal approximator $f$ (e.g., neural network) is essentially to learn $f$ independently in each environment/domain, which does not make use of the prior information **there is some invariance across domains** at all.
> > > > >
> > > > > ### Response to 2.
> > > > >
> > > > > The proposed method **IRM-ACDM outperforms the EIIL consistently** in terms of both (higher) mean and (lower) standard deviation in test accuracy.
> > > > >
> > > > > ### Response to 3.
> > > > >
> > > > > This reviewer’s claim 3. is not correct. If possible, could the reviewer point out which part of [3] supports the claim? Generally speaking, $P(Y|X^c)$ is invariant across domains is not a sufficient condition for $X^c$ to be exactly the set of causes for $Y$. For the simplest example, $X^c$ can be just a subset of the causes of $Y$ which are exogenous. In fact, in traditional causal inference, even if we know $X^c$ is a cause of $Y$, we do not have information about whether $P(Y|X^c)$ is invariant across domains. One example is a non-stationary system [2] where $P(Y|X^c)$ can change over time even if $X^c$ is exactly the set of causes of $Y$.
> > > > >
> > > > > [1] Arjovsky, Martin, Léon Bottou, Ishaan Gulrajani, and David Lopez-Paz. "Invariant risk minimization." arXiv preprint arXiv:1907.02893 (2019).
> > > > >
> > > > > [2] Huang, Biwei, Kun Zhang, Mingming Gong, and Clark Glymour. "Causal discovery and forecasting in nonstationary environments with state-space models." Proceedings of machine learning research 97 (2019): 2901.
> > > > >
> > > > > [3] Bareinboim, E., Brito, C., & Pearl, J. (2012). Local characterizations of causal Bayesian networks. In Graph Structures for Knowledge Representation and Reasoning (pp. 1-17). Springer, Berlin, Heidelberg.

---

> > > > > > ### Comment · AnonReviewer3 · 2020-11-24
> > > > > > **Response**
> > > > > >
> > > > > > Hi
> > > > > >
> > > > > > 1. You are missing the point I am trying to make here. In the simple system with $Y_e \leftarrow f(X_e) + n_e$, with some additional regularity conditions, it can be shown that $f()$ is the ideal invariant model one should learn. Do you agree to this or not? Note I am not making any assumption of overlap or not. $f()$ can map to linear models as in Theorem 9 or non-linear models. Please read IRM carefully. You can think of the whole ideal representation to be a $f()$ and consider a simple linear function on top $w$. So the composite model is $w.f()$ and $w$ is set to $1$. This is exactly what IRM does.
> > > > > >
> > > > > > 2. So if you agree that $f()$ is the ideal representation that you are trying to learn. It does not seem true that $f()$ will minimize a regularized CDM as well for any value of regularizer. If your claim is you can prove that $f()$ minimizes a regularized loss for certain $\beta$, then please show a proof. Your **ideal model** should at least be captured exactly as a solution to some instance of your **optimization** or else it is a **heuristic**.
> > > > > >
> > > > > > 3. Even if you do not want to learn the ideal representation $f()$ in the way I described. I want to ask you a simple question--can you at least for a large class of SEMs show what the ideal representation and classifier on top that you want to learn and then go on to show that it minimizes the criterion that you have for some value of regularization as you claim.
> > > > > >
> > > > > >
> > > > > > 4. Now having made the larger point. In the paper you never introduced regularization with this goal. You only stated regularization as a means to solve the constrained problem. You never stated regularization was meant to circumvent the problem I am talking about.
> > > > > >
> > > > > > 5. In the paper that I cited, all I am saying is suppose you are dealing with Causal Bayesian Networks (CBN). In such a case, the local view on invariance given by the modularity condition is equivalent to defining a CBN (See Theorem 1 and Definition 4).

---

> > > > > > > ### Author Response · Authors · 2020-11-25
> > > > > > > **Thanks for the reviewer's response! We justify our disagreement with the reviewer's claims.**
> > > > > > >
> > > > > > > 1. The claims made by our paper are (1) IRM suffers from the non-overlapping issue of $F(X)$ from different domains under strong $\Lambda$ spuriousness with deep models, and (2) IRM-CDM is a simple and effective fix to this non-overlapping issue. We empirically show that the proposed IRM-CDM improves IRM on CMNIST+. **We never claim that it has theoretical guarantees on recovering some-sort of causal representations, for example, the causal representation defined by a SEM. However, the reviewer wrote: *If your claim is you can prove that $f()$ minimizes a regularized loss for a certain $\beta$, then please show a proof.* It may not be okay for the reviewer to first make an assumption for our work and then to attack the assumption made by him/herself.**
> > > > > > >
> > > > > > > 2. Every method has its assumptions and domains to work, so does the proposed IRM-CDM. IRM-CDM is a generalized version of the vanilla IRM. IRM can be treated as IRM-CDM with $\beta = 0$. IRM-CDM is applicable to the cases where IRM is supposed to be applicable. Our results show solid improvements over IRM in the cases where IRM suffers from the non-overlapping issue. We agree that it is an important problem to theoretically quantify classes of problems where the proposed method has theoretical guarantee.  We leave it as a future work. We welcome the reviewer to propose **concrete examples** that the method fails (there exist such examples) and we would love to explicitly mention those cases in the next version of the paper. However, based on the reviewer’s, we have not seen such a formally defined class of problems that IRM-CDM fails. The reviewer first proposed the linear model in Theorem 9 [1]. But later we agreed that it does not apply to the deep invariant feature learning setting that this paper focuses on. After that, **the reviewer only wrote the following argument, which lacks formal specification and contradicts with the problem setting of IRM and this work. The reviewer wrote: In the simple system with $Y_e←f(X_e)+n$, with some additional regularity conditions, it can be shown that $f()$ is the ideal invariant model one should learn. Do you agree to this or not?** In IRM, the goal is to learn a representation $F(X)$ that elicits an invariant optimal predictor for all domains [1]. This implies that it is unlikely, in practice, that we can learn the invariant function $f$ that directly maps raw features $X_e$ to the label $Y$, even if it exists. Instead, we need to learn a feature representation $F(X)$, which captures the *causal features* that elicits an invariant predictor mapping $F(X)$ to the label $Y$ across domains. In addition, in the reviewer’s question, he/she did not specify (1) what is the *simple nonlinear system* or (2) what are the *additional regularity conditions*. If the reviewer can provide a formal definition of the aforementioned terminologies along with the derivation to support his/her claim since the reviewer wrote **it can be shown ...**.We cannot agree or disagree with the reviewer when a solid formulation of the reviewer’s claim is absent.
> > > > > > >
> > > > > > > 3. Please note that **the regularization formulation of IRM-CDM is the method proposed in both the first and current version of our paper.** We will try to improve the presentation of the motivation since the reviewer seems to misunderstand it.
> > > > > > >
> > > > > > >
> > > > > > > [1] Arjovsky, Martin, Léon Bottou, Ishaan Gulrajani, and David Lopez-Paz. "Invariant risk minimization." arXiv preprint arXiv:1907.02893 (2019).

---

> > > > > > > > ### Comment · AnonReviewer3 · 2020-11-25
> > > > > > > > **Response**
> > > > > > > >
> > > > > > > > Hi
> > > > > > > >
> > > > > > > > Thanks for your response. I have read all your responses carefully and I still disagree with your claims.
> > > > > > > >
> > > > > > > > 1. **About the regularity conditions and why f is indeed the ideal solution:**
> > > > > > > >  Please read Section 4 page 10 and 11 of the IRM paper [Arjovsky et al.]. It is stated that the function $v(x) = \mathbb{E}[f(Pa(Y), N_Y)]$ solves the OOD problem. In the example I gave you, if noise is zero mean and has bounded variance, then $v(x)= f(x)$.  Now if you want detailed proofs of how $v()$ solves the OOD problem, please refer to Theorem 3.6 in https://arxiv.org/pdf/2008.01883.pdf.  I hope you are now convinced that $f()$ solves the OOD problem.
> > > > > > > >
> > > > > > > >
> > > > > > > > 2. **About counter-example to your work**:  Before getting into the details,  if a work claims not to be a heuristic then it has to have some ground in theory. In your case that would mean that providing some examples when the the method works in theory under some ideal assumptions. You have not provided any convincing example.   I have already provided some good amount of examples why your method does not work.
> > > > > > > >     a) At first, I gave you a simple linear model as an example as it is clearly stated in the work of Arjovsky (I had clearly mentioned in my original response this was for the sake of discussion and not meant to be thought that it only applies to linear models).  I had described why the linear model example serves as a counter to your CDM approach.  To this you claimed your setup works only for non-linear models. When there is no theory in the paper, how is one supposed to discern if it works for linear models or non-linear models? There were no discussions on this either in the paper.
> > > > > > > >
> > > > > > > >     b) Next, I stated the non-linear generalization example. On hearing this example, you said that my argument is based on the constrained version of the problem and your method is designed for the regularized case.  Let me remind the authors that the  justifications in your paper before equation (3) are based on the constrained setting.
> > > > > > > >
> > > > > > > >     c) Lastly, on reading about the regularized case, I asked you to provide a simple set of examples that satisfy the following
> > > > > > > >
> > > > > > > >    i) Satisfy lambda spuriousness
> > > > > > > >     ii) Follow some generative model, which will help you identify the target ideal model you want to learn
> > > > > > > >     For such a family you can choose a simple linear or non-linear family whatever you like and should show that the target model solves the regularized CDM problem.
> > > > > > > >
> > > > > > > > In your response you say "We never claim that it has theoretical guarantees on recovering some-sort of causal representations, for example, the causal representation defined by a SEM." I do not have a problem if you do not have any theoretical justification for your method in the paper. Please realize that I have a problem with the fix proposed in the paper and I have provided enough discussion and examples for the same. Even if the fix works for one dataset I do not think that is sufficient grounds to think that the method actually works more generally.
> > > > > > > >
> > > > > > > > To conclude, I would request the authors to consider the many suggestions I have made after spending a considerable amount of time on reviewing your work.

---

> > > ### Author Response · Authors · 2020-11-23
> > > **Response to (c) (results of EIIL on CMNIST+) and (e) and a kindly request to reevaluate our work**
> > >
> > > We thank the reviewer for the insightful suggestions. We really appreciate the time and effort you paid in the review.
> > >
> > > ### Response to (c): Results of EIIL
> > > We evaluate EIIL [3] (https://github.com/ecreager/eiil) on CMNIST+. EIIL gets test accuracy $43.40 \pm 10.32\%$, $43.24 \pm 13.03 \%$, $40.93 \pm 13.03 \%$ on CMNIST+ with $\rho=0.8,0.85,0.9$. Compared to IRM-ACDM $57.23 \pm 4.34\%$, $44.83 \pm 4.65 \%$, $42.85 \pm 3.09 \%$ and IRM-MMD $52.91 \pm 4.56\%$, $40.83 \pm 2.50 \%$, $37.96 \pm 6.97 \%$, EIIL has comparable mean accuracies when $\rho = 0.85,0.9$, but is not stable. There are two reasons. (1) EIIL relies on IRM and the soft domain weight $q(E|X,Y)$. When $q(E|X,Y)$ takes values s.t. strong $\Lambda$ spuriousness exists, it makes IRM fail. (2) It has issues in model selection. EIIL can only select models by validation accuracy as $q(E|X,Y)$ is a scalar $q_i$ for instance $i$. It is not learned for the validation or test, so validation loss is not computable. Note that validation accuracy can be high when the model picks up spurious features. In contrast, model selection with validation loss considers the regularizers (IRM and CDM), which approximates how well models fit causal features. We add these results in Section 4 and Appendix C3.
> > >
> > > ### Response to (e): The data generating process (DGP) of CMNIST+ and causal graph (Fig. 1).
> > >
> > > The DGP is as follows:
> > >
> > > 0. In MNIST, shapes $X^c$ decide digit labels.
> > > 1. We get true labels $Y^*$ of instances by digit labels (0-9).
> > > 2. We split the data into test and training.
> > > 3. We assign the training instances to the training domains based on $Y^*$ and $P(Y^*|E)$ in Table 1. This introduces correlations between $Y^*$ and E. In training domains, we split data into training and validation.
> > > 4. We generate noisy labels $Y$ by randomly flipping them with 25% probability.
> > > 5. Given $P(C|Y,E)$, the $Y$ and $E$, we assign colors. This step introduces correlations among $X^s$, $Y$ and $E$.
> > >
> > > There is a difference in what causal relationships mean in traditional causal inference and in OOD prediction. In OOD prediction, the definition of causal relationships is different from a traditional one. Traditionally, $X^c \rightarrow Y$ means the generation of $Y$ is influenced by $X^c$. It does not necessarily mean $P(Y|X^c)$ remains the same across domains [2]. However, in OOD prediction, we say there exists a causal relationship $X^c\rightarrow Y$ iff $P(Y|X^c)$ is the same across different domains.
> > > In the DGP of CMNIST+, (1) there exist causal relationships $X^c\rightarrow Y^* \rightarrow Y$, so $P(Y|X^c)$ is invariant across $E$. (2) There are correlations among $X^s$, $Y$ and $E$.
> > > To let Fig.1 be in accordance with the DGP, we make the directed edges among $E$, $X^s$ and $X^c$ to be bidirected to signify correlations. We describe the DGP in Appendix C2.
> > >
> > > ### Summary
> > > Our response covers the reviewer’s concerns from (a) all the way to (g). We also updated the manuscript accordingly. We believe we resolved the concerns and hope the reviewer can kindly reevaluate our work.
> > >
> > > [1] Arjovsky et al. "Invariant risk minimization." arXiv preprint arXiv:1907.02893 (2019).
> > >
> > > [2] Pearl and Bareinboim. "External validity: From do-calculus to transportability across populations." Statistical Science (2014): 579-595.
> > >
> > > [3] Creager et al. "Environment Inference for Invariant Learning." In ICML Workshop on Uncertainty and Robustness. 2020.

---

### Official Review · AnonReviewer1 · 2020-10-27
**This paper discusses an interesting problem of invariant causal feature learning applied in OOD learning scenarios. It proposes to use a divergence based regularization term to address the spurious correlation existing in IRM.**

**Rating:** 6
**Confidence:** 3

**Review:**

This paper attacks the problem of OOD learning from the angle of invariant causal feature learning. The key idea is to capture domain invariant causal features and use the extracted causality relation to convey domain-adaptive classification. In this work, domain invariant causal features are learned by IRM, which imposes the consistency constraint between causal features and class labels across different domains. The core idea is to address the existence of spuriousness correlation by introducing the MMD and KL divergence based conditional distribution matching constraint to the IRM learning process. The experimental study based on a crafted MNIST data set demonstrates the superior performances of the regularised IRM learning method in the domain invariant learning task.

In general, the paper introduces an in-depth discussion of the limitation of the IRM learning mechanism and points out the root cause of failure of IRM (spuriousness correlation). This is interesting and potentially impactful for practical OOD learning tasks. The paper is well-written and the proposed objective is novel to my knowledge. we tend to accept the paper.

Still, our concerns are as follows:
1. Though the results look promising on the toy data set, it would be better to have a real-world scenario as a testbed for the proposed method. Domain transfer is a popular application. How would this method perform in a domain transfer learning task?
2. Following the first question, we would expect some discussion about the relation between the proposed method and other transfer learning methods, such as meta-learning methods. Could domain invariant casual feature learning be considered as a way of conducting meta-learning?
3. A minor issue in Table.1: how many domain labels are there defined in the CMNIST data set? How are they defined?

---

> ### Author Response · Authors · 2020-11-16
> **About evaluation, application to domain transfer learning and meta learning.**
>
> We thank the reviewer for the thoughtful suggestions and detailed reviews.
>
> We agree it would be better if we can evaluate IRM-CDM against IRM, CDM and ERM in a real-world scenario. However, it can be challenging to confirm that there exists strong $\Lambda$ spuriousness in a real-world dataset. We will try to add such an experiment in the next version of this work.
>
> In this paper, we specifically focus on the limitation of IRM under strong $\Lambda$ spuriousness instead of general domain transfer learning. The expected performance of the proposed method in domain transfer learning tasks can vary by the problem settings. Our method is specifically designed for cases where the causal relationships $P(Y|X^c)$ is invariant while statistical associations $P(Y|X^s)$, $P(X^c|E)$ and $P(X^s|E)$ can change across domains.
>
> We also conjecture that invariant causal features can be useful in meta learning tasks under proper conditions. Since this topic is not very relevant to our paper, we would like to let the reviewer find answers in [1,2].
>
> As shown in Table 10 in Appendix B.2, the original CMNIST dataset has two training domains and a test domain. The detailed setup ($P(Y|E)$ and $P(C|Y,E)$) can be found in Table 10.
>
> [1] Zhang, Marvin, Henrik Marklund, Abhishek Gupta, Sergey Levine, and Chelsea Finn. "Adaptive Risk Minimization: A Meta-Learning Approach for Tackling Group Shift." arXiv preprint arXiv:2007.02931 (2020).
> [2] Yue, Zhongqi, Hanwang Zhang, Qianru Sun, and Xian-Sheng Hua. "Interventional few-shot learning." Advances in Neural Information Processing Systems 33 (2020).

---

### Official Review · AnonReviewer2 · 2020-10-28
**Valuable study of and improvement over invariant risk minimization**

**Rating:** 7
**Confidence:** 4

**Review:**

### Summary of Paper

This paper identifies and tries to fix a limitation of the recent work of Invariant Risk Minimization (Arjovsky et al., '19). IRM is a solution framework for the OoD prediction problem, where one has to learn a classifier based on data from multiple domains, hoping to generalize to unseen domains.

1. This paper extends the Colored-MNIST dataset (where MNIST images are spurious colored according to the label) to a new and more difficult dataset called CMNIST+ where they empirically show that IRM fails to generalize to the new domain. The key idea behind the construction of the CMNIST+ dataset is the introduction of a correlation between the spurious features and the _domain_ label during training time (besides the training-time correlation between the spurious features and the class label that is already there in Colored-MNIST). This spurious-feature-domain-label correlation can be introduced by making Domain 1 largely contain points of class 1 and Domain 2 largely contain class 2. They call this sort of correlation as "$\Lambda$-spuriousness".

2. The paper provides an intuitive argument for why IRM does not work on this dataset: the IRM constraint is equivalent to "class label independent of domain label given feature representation", such a constraint does not preclude the classifier from learning "feature representation = domain label". Such a classifier however would not generalize well to an unseen domain where the domain label is not correlated with the class label.

3. Finally, the paper proposes a fix for IRM which essentially adds a "conditional distribution matching" constraint to the IRM constraint. This constraint forces the distribution of the feature representation for any class label to be invariant across domains. By implementing this through (a) an MMD approach and (b) an adversarial learning approach, they show a 10% improvement in OoD accuracy on CMNIST+.

### Strengths

1. The paper is strong on novelty: the problem identified, the explanation provided, the dataset proposed, and the solution proposed are all novel. Overall, the paper piqued my curiosity and I enjoyed reading it.

2. The problem of OoD prediction is practically important. Furthermore, this paper exposes the limitations of an existing solution under a very natural kind of spurious correlation that could occur in real life (i.e., domain-class correlation).

3. The paper also provides a synthetic dataset that will be quite valuable to future work that aims to develop better OoD prediction algorithms.

4. The "failure" dataset and the solution proposed are all founded on a solid, intuitive argument.


### Weaknesses + important clarification questions

5. I think the paper will benefit greatly from at least one other empirical example both in terms of showing failure of IRM and in terms of showing that IRM-MMD/ACDM improves. This could either be on a synthetic dataset similar to (but not) CMNIST+  (maybe MNIST but with some other spurious feature; or maybe some other dataset with the same spurious feature). Even better, it'd be great (but not absolutely necessary) to verify the performance of IRM-MMD/ACDM on a practical benchmark.


6.  I was confused about the definition of $\Lambda$-spuriousness. The introduction defines this to be the existence of a correlation between label and color during training. Isn't this the same correlation that exists in CMNIST? Is this a typo? I suspect that the $\Lambda$-spuriousness refers to strong correlation between the class label, the domain label and the spurious features. Or did I completely misunderstand this?

7. Could you explain why you had to resort to using three channels/colors as against just two like in CMNIST? I wonder if this is the point that was addressed by the following line in the paper:
> The Colored MNIST (CMNIST) dataset cannot expose the limitation of IRM under strong Λ spuriousness. This is because its two training domains are quite similar.

**Update:**   You provided an example 2-color dataset to argue why you can't really see whether IRM fails in the test domain. However, this example doesn't seem to be the correct analog of the example 3-color dataset in your paper? In your 2-color example, domain 1 is mostly Y=1, and most of those datapoints are colored G. Domain 2 is mostly Y=0 and most of those datapoints are colored G too. But the analog of the 3-color dataset would be one where in domain 2 most datapoints have Y=0 and those datapoints are mostly colored B (or R, but not G). Then, consider a test domain where you've an equal proportion of Y=0 and Y=1, but all Y=0 are colored G and Y=1 are colored by  B.  It seems like under this case, if IRM were to use the color, it would have a poorer test accuracy than 0.5. Perhaps I'm missing something here. Nevertheless, it seems like considering a dataset like this, and/or fleshing out a corresponding Table 1 for such a dataset would be critical to substantiate this argument.





8. The argument that label balancing does not fix IRM's issue is a crucial argument to justify the fix given here. But I would have appreciated a bit more elaboration on this. The text says "Theoretical analysis shows that this [label balancing] is an invalid solution". Could you explain how I can infer this from Table 2? Which column here would correspond to the performance of IRM under label balancing, and why?


### Overall opinion
The paper identifies a novel gap in an existing algorithm for an important problem, provides an intuitive explanation as to why that gap exists, and also proceeds to provide a reasonable, intuitively-grounded fix for it. Overall, this makes a complete, coherent paper worth publishing.

#### Minor suggestions
- For completeness, it would be nice to provide a concrete argument and discussion (in the appendix) for the connection between the original IRM constraint and the constraint ``"Y \condindep E | F(X)"
- In Page 3, there is a footnote against the symbol "E" which might be better positioned elsewhere.
- In Fig 2 (c), $K_{IRM} = 0$ could be misleading (since it can be interpreted as adding the constraint right from the first step). Perhaps $K_{IRM} = -1$ or $K_{IRM} = \infty$ would be more appropriate.


#### References

Martin Arjovsky, Le ́on Bottou, Ishaan Gulrajani, and David Lopez-Paz. Invariant risk minimization.
arXiv preprint arXiv:1907.02893, 2019.


**Update**: Thanks to the authors for clarifying most of my clarification questions.  I'm not sure I was able to fully follow your argument about why these results can't be adapted to a two-color dataset (see above), but to indicate that you've clarified many of my questions I've increased my confidence score to a 4.  Good luck to the authors.


**Further updates**: I'd like to elaborate on my thoughts a bit more with the hope that the authors may find it useful for future versions of the paper.

- First, I really appreciate the authors for performing additional experiments with EIIL during the response phase. I wish to emphasize that, after a long discussion with R3, I've some strong disagreements with their review regarding the "simple fix":
   -   I personally think EIIL is out-of-scope as it is a recent algorithm. It doesn't sound like a "simple fix" to me. However, it's great that the authors were able to show that their algorithm works better.
   -  I don't think pooling all datapoints and splitting it will work. You'll end up with a dataset with label-color correlation in both domains, and both domains will be identical. So IRM won't work here.
  - One can always come up with some hacks like "pool all environments and carefully split them back" that work under the assumption that there's $\Lambda$-spurious correlation. But those hacks would be sub-optimal if there were no $\Lambda$-spurious correlation.  Therefore, this is an unfairly powerful "overfit" hack, and does not make a good baseline. You want an elegant solution that works whether or not there's $\Lambda$-spurious correlation as you won't know whether that sort of a spurious correlation exists in practice.
  - As a side note, in light of the above point, I think it's important that the authors also demonstrate that the CDM constraint added preserves the performance of IRM on the original CMNIST dataset.
  - Hopefully the authors can keep the EIIL results for future versions of the paper as it only makes the paper stronger.

- I don't think the paper should be heavily penalized for the lack of a realistic dataset, because it's hard to verify $\Lambda$ spuriousness on realistic benchmarks. However:
  - I'd strongly encourage that you consider trying similar experiments on a dataset like say Rotated-MNIST (or Rotated-MNIST+ to be more precise, if at all possible).
  - Even better, you could consider whether similar experiments can be done with Celeb-A where you have access to image attributes like hair color etc., (See Fig 2 https://arxiv.org/abs/2005.04345) and you could try creating different environments by sampling differently in each.
  - If you think that's it's impossible to create a $\Lambda$-spurious dataset, you might want to explain in future versions of the paper as to why that's not possible.

- I understand R3's main concern which is that the algorithm in this paper requires that the distribution of the causal features $X_{causal} | Y$ to be the similar across all environments. It seems like IRM doesn't expect this sort of invariance, while algorithms prior to IRM do require something of this sort (including CDM, DANN etc.,). I think one actionable way to address this concern would be to show that there are datasets where IRM + CDM does better than CDM (just like how IRM+CDM does better than IRM in CMNIST+). This way we can see why combining IRM and CDM offers something unique.


- Finally, I want to appreciate your efforts in trying to clarify all the reviewers' concerns (at least I found them helpful) and also to update the paper accordingly, add experiments etc.,

---

> ### Author Response · Authors · 2020-11-18
> **We thank the reviewer for the helpful suggestions and thoughtful reviews**
>
> ### More benchmarks
>
> We agree that adding more benchmarks can improve our confidence in IRM-CDM’s effectiveness in solving the OOD prediction problem under strong $\Lambda$ spuriousness. As it can be extremely challenging to confirm that there exists strong $\Lambda$ spuriousness in a practical benchmark, we plan to add another semi-synthetic benchmark in the next version of this work.
>
> ### Λ spuriousness
>
> As we introduced in the third paragraph of introduction, the strong Λ spuriousness means the strong spurious correlation between $X^s$ (spurious features) and $Y$ (label) is via their common cause $E$ (domain variable). This means the causal effect $E$->$X^s$ and $E$->$Y$ are both strong. This is different from the scenario of CMNIST. In CMNIST, since the two training domains are similar to each other, the causal effect $E$ -> $X^s$ is weak. Therefore, the strong Λ spuriousness does not exist in CMNIST. We will update the manuscript to make it clearer.
>
> ### Three colors in CMNIST+
>
> The main reason to use three colors is that it is not easy to create a proper benchmark for OOD prediction under strong $\Lambda$ spuriousness with only two colors. As shown in Figure 3, we can see with two colors, we can only create domains on the line between R (red) and G (green). So, it would be either (1) the two training domains are very similar to each other (as it is in CMNIST) or (2) the test domain is similar to one of the training domains. In case (1), strong $\Lambda$ spuriousness does not hold. In case (2), it is generally not ideal to benchmark OOD prediction since we expect invariant causal features should let the model generalize to unseen test domains which are not similar to the training ones.
>
> ### Theoretical results (Table 2) regarding IRM’s performance with label balancing
>
> To read Table 2 regarding IRM’s performance with label balancing, we first need to understand that the conditional independence of IRM ($Y \perp E | F(X)$) can be satisfied by E, Concat(E,C) and S, so we should look at the three corresponding columns on the right half of Table 2. We know that the model resulting in *best training accuracy* would be learned among the three. As $\rho$ increases, we should expect IRM with label balancing is expected to be more likely to pick up Concat(E,C) as its feature representation, which leads to test accuracy = 0.2. In practice, we can see as $\rho$ increases, the test accuracy of IRM with label balancing becomes closer to 0.2. We will add a paragraph to clarify the connection between Table 2 and Figure 1.
>
> ### Minor suggestions
>
> We will update the manuscript with (1) a clarification on the relationship between IRM and $Y \perp E | F(X)$. You can also find our answer in the first part of our response to reviewer 3. (2) We will move the footnote to another place and (3) make the meaning of $K_{irm}$ easier to understand.

---

> > ### Comment · AnonReviewer2 · 2020-11-20
> > **Clarification about two colors**
> >
> > Thank you for your responses! I'm going to reorder my concerns in terms of their importance, based on your clarifications.
> >
> > **Three colors in CMNIST+**: I'm afraid I'd still like some clarification here because it seems like I'm missing something. I agree with your point about case (2) being uninteresting and with case (1) in that in the original CMNIST dataset the two training domains are very similar to each other. But this doesn't clarify why one can't create a two-color dataset with strong $\Lambda$ spuriousness.
> >
> >  That is, imagine a variation of CMNIST where the first domain contains mostly points from class 0 and the second domain contains mostly points from class 1. Imagine that in both domains, class 0 is most likely to be red and class 1 most likely to be green. Wouldn't IRM still potentially fail here? By looking at the color of the datapoint $X$, the predictor could predict the domain label as its representation $F(X)$, and hence you may still have that $Y$ is independent of $E$ given this representation. So essentially, my question is
> > - can you create strong $\Lambda$ spuriousness with just two colors during training and testing?
> > - if you can create it, does IRM fail here? Why/why not?
> >
> > Clarifying this would be important in order to motivate why we need to think about CMNIST+.
> >
> >
> > **More benchmarks**: I understand. There are no realistic benchmarks where we can examine and play around with the spurious features. To clarify, the absence of a realistic dataset doesn't affect my score.
> > I do think another synthetic dataset would be valuable to add.
> >
> >
> > **$\Lambda$ spuriousness**: Thanks for the clarification. Since you say "As we introduced in the third paragraph of introduction", I must say that I did read this paragraph multiple times. Perhaps, it'll be helpful to state which two lines in this paragraph confused me: "Specifically, we consider the situation where strong spurious correlation exists between the spurious features and the class label only in the training set. We name this strong Λ spuriousness..." The first line doesn't make any reference to the domain label at all, so it makes it seem as though any kind of strong correlation between the two can be termed $\Lambda$ spuriousness. Of course, the rest of the paper clarifies this, but I think these two lines can be rewritten, perhaps more formally. i.e., explicitly state what's going on in Fig 1 in that paragraph. Anyway, I acknowledge that this is just a minor presentation issue at this point.
> >
> >
> > **Table 2**: Thanks, that makes sense!
> >
> > Finally, I'd like to encourage you to update the paper with whatever changes you can make based on this feedback (including those on the presentation) before the rebuttal period ends. Thanks again for your clarifications!

---

> > > ### Author Response · Authors · 2020-11-23
> > > **Three colors in CMNIST+, Λ spuriousness and updated manuscript.**
> > >
> > > We really appreciate the reviewer's insightful and helpful suggestions!
> > >
> > > ### Three colors in CMNIST+
> > >
> > > #### Q1. Can we create strong $\Lambda$ spuriousness with just two colors during training and testing?
> > >
> > > A1. It is possible to create strong $\Lambda$ spuriousness with just two colors for binary classification with two training domains. Let’s say we use red (R) and green (G). To show this is possible, we use an example with the following setup: P(Y=1|E=1)=0.9, P(Y=1|E=2)=0.1. Let’s say for $E=1$, we set $P(C=G|Y=1,E=1)=0.9$, $P(C=G|Y=0,E=1)=0.1$. Then, we can set $P(C=G|Y=1,E=2)=0.1$, $P(C=G|Y=0,E=2)=0.9$ for $E=2$. We can observe strong correlations between color, class label and domain label. Thus, we created strong $\Lambda$ spuriousness with just two colors.
> > >
> > > #### Q2. Would IRM still fail in a two-color datasets with strong $\Lambda$ spuriousness?
> > >
> > > A2. As our discussion on why IRM fails and why the proposed fix is effective does not depend on the number of colors in a dataset. Our concern with such datasets is that even if strong $\Lambda$ spuriousness exists in training domains, it is a challenge to find a test domain that is quite different from both training domains. Following the example in A1., for the test domain $E=3$, if we set $P(Y=1|E=3)=0.5$, $P(C=G|Y=1,E=3)=0.5$ and $P(C=G|Y=0,E=3)=0.5$, it would be right in the middle of the two training domains. In this case, even if IRM fails, it can be difficult to show it with the test accuracy. A model only fits colors can reach 0.5 accuracy, which would show smaller differences between models fitting causal features and those fitting the spurious ones. This would make it more challenging to judge whether a model failed in an experiment.
> > >
> > > We added these explanations to Appendix C2.
> > >
> > > ### Λ spuriousness
> > > Thanks for pointing out the confusion! We rewrote the sentence as “Specifically, we consider the situation where strong spurious correlations among the spurious features, the class label and the domain label only hold for training data.” Hope this makes sense to you.
> > >
> > > ### Updated Manuscript
> > > Thanks for your suggestion! We updated the manuscript according to the review! We hope it is helpful for you to better understand our work.

---

### Official Review · AnonReviewer4 · 2020-10-28
**Nice touch, but justifications need to be substantially enriched**

**Rating:** 4
**Confidence:** 4

**Review:**

**Summary.** This paper advances generalizable machine learning via addressing a major limitation of invariant risk minimization (IRM). In particular, the author(s) identified and discussed the issue of strong $\Lambda$ spurious, where spurious features and class labels are strongly correlated due to common cause, causing unprotected IRM to fail while trying to exclude such non-causal predictors. To avoid this. pitfall, the author(s) proposed to leverage conditional distribution matching (CDM) to regularize the representation, which effectively helps to alleviate this issue. Two empirical solutions, respectively non-adversarial MMD and adversarial KL matching, have been presented and validated.

**Quality & Clarity.** Overall this paper is presented with clarity. The problem is well motivated and carefully discussed. What I found unsatisfactory is the proposed solution needs extra justifications, which is detailed in my weakness section below.

**Originality & Significance.** The author(s) have identified a major weakness of IRM that I also find concerning: while developed from the notation of invariant representations, based on which invariant predictors are defined, IRM does not explicitly regularize the representation in its formulation. This view, provided fully developed, encapsulates sufficient novelties. On the significance side, while this submission indeed addresses a major concern of IRM demonstrated by artificial examples, the author(s) fail to present a concrete real-world example to showcase this concern is a thing that we should actually worry about.

**Main Weakness.**
Justification of CDM needs to be strengthened. The author(s) have provided an argument that explains the extreme case, where non-causal features perfectly predict domain. The discussion needs to be substantially enriched. CDM has been proposed for dealing with the label shift in domain adaption, and it relies on assumptions that should be reconciled with those made by IRM. Please clarify.

Theoretical results on pp 4. This is a totally misleading heading. What I have expected is some theoretical discussion, instead, the author(s) have provided a numerical table computed from "theoretical computations". I do not consider these as theoretical as they do not generalize beyond this particular example.

Insufficient experimental validation. This is what kills this paper. There is only one experiment performed on a semi-synthetic testbed, which does not serve to evidence the practical utility of this proposal.

The domain adversarial neural net (DANN) model is highly relevant to the proposal made here and should be carefully discussed and compared. In fact, DANN also regularizes the representation to make it domain-agnostic.

**Minor issues.**
The term out-of-distribution (OOD) is a bit misleading, as this phrase is usually associated with the task of anomaly detection, where novel samples that are very different from the training examples are identified.  I would suggest the author(s) replace OOD to avoid confusion.

The aspect ratio in Fig 1 is off and it. makes readers very uncomfortable. Please redo this figure. And it does not clearly depict what's different compared to the standard scenarios amendable to IRM.

---

> ### Author Response · Authors · 2020-11-16
> **Justification on CDM, Theoretical results on pp 4, Insufficient experimental validation and Comparison with DANN**
>
> ### Justification on CDM
>
> For OOD prediction under strong $\Lambda$ spuriousness, IRM-CDM can be justified by (1) the IRM constraints can lead to undesired feature representations that (partially) fit the domain label and (2) CDM can compensate for IRM to make it more difficult to (partially) fit the domain label. In addition, considering an invertible function $f$ as the invariant classifier that maps $F(X)$ to $Y$, then given the label $Y=y$, through $f^{-1}$, we should expect to find the feature representations of two instances from the same class to be sampled from the same distribution $P(F(X)|Y)$ without knowing which domains they are from. Note that, when it is not the extreme case, having information of the domain label in the feature representations is also undesired as $P(Y|E)$ is not an invariant relationship across domains.
>
> ### Theoretical results on pp 4
>
> We agree that our “theoretical results” are not the traditional type of theoretical results. We will replace “theoretical results” with more proper terminology. However, the way we perform the theoretical analysis can be extended to other datasets.
>
> ### Insufficient experimental validation
>
> We agree that it would be better if we add more benchmarks. However, we must argue that the core arguments of this paper are that (1) IRM fails under strong $\Lambda$ spuriousness as it can be satisfied by fitting the domain label and (2) IRM-CDM can fix the problem of IRM in this scenario as it can exclude undesired solutions that fit the domain label. The experiments included in the paper considering various strengths of $\Lambda$ spuriousness have already shown enough evidence to support our arguments.
>
> ### Comparison with DANN
>
> DANN is a method to align feature representations across domains, which imposes unconditional distribution matching, i.e., $P(F(X)|E) = P(F(X)|E’) = P(F(X))$. Unconditional distribution matching is not a solution to OOD prediction as it fails when P(Y|E) varies across domains.
>
> ### Minor issues
>
> We understand that OOD is also used in the anomaly detection literature, but in causal machine learning, OOD is widely used to refer to the problem we discuss in this paper [1,2].
> We will update Fig. 1 to make it easier to depict and understand the difference between strong $\Lambda$ spuriousness and the scenario of the original CMNIST.
>
>
> [1] Arjovsky, Martin, Léon Bottou, Ishaan Gulrajani, and David Lopez-Paz. "Invariant risk minimization." arXiv preprint arXiv:1907.02893 (2019).
>
> [2] Krueger, David, Ethan Caballero, Joern-Henrik Jacobsen, Amy Zhang, Jonathan Binas, Remi Le Priol, and Aaron Courville. "Out-of-distribution generalization via risk extrapolation (rex)." arXiv preprint arXiv:2003.00688 (2020).

---

### Author Response · Authors · 2020-11-24
**Response and revision**

We thank reviewers for their insightful feedback and for appreciating our work! We have revised the paper with the following major changes to incorporate the comments.

Based on R2’s suggestions, we add a paragraph in Appendix C.2 to explain why using two colors to set up a colored MNIST dataset under strong $\Lambda$ spuriousness is possible but not ideal.

Based on R3’s suggestion, in Section 4, we add experimental results of the baseline EIIL [1] on CMNIST+, which show EIIL suffers from large standard deviation even if it can achieve comparable mean test accuracy to the proposed method IRM-CDM. We also provide a detailed explanation why it is unstable in Appendix C3.

Based on R3 and R4’s suggestions, in Appendix A, we add a detailed description of the relationship between IRM and the conditional independence $Y \perp E | F(X)$.

Based on R3’s suggestions, we add a detailed description of the data generating process of CMNIST+ in Appendix C.2. We also update the causal graph in Fig. 1 to make sure it is well aligned with the data generating process of CMNIST+. In Appendix C.2, we explain why they align with each other.

Based on R2’s suggestions, we rewrote the confusing sentence *Specifically, we consider the situation where strong spurious correlation exists between the spurious features and the class label only in the training set. We name this strong Λ spuriousness...* in the third paragraph of Section 1 as *Specifically, we consider the situation where strong spurious correlations among the spurious features, the class label and the domain label only hold for training data.*

Based on R4’s suggestion, we changed *theoretical results* to *analysis results* in Section 3.1 and Appendix B as they are specific for the CMNIST+ dataset.

[1] Creager, Elliot, Jörn-Henrik Jacobsen, and Richard Zemel. "Environment Inference for Invariant Learning." In ICML Workshop on Uncertainty and Robustness. 2020.

---

### Decision · Program_Chairs · 2021-01-07
**Final Decision**

**Decision:**

Reject

**Comment:**

Loosely, while IRM aims to find a feature mapping Phi s.t. response Y given Phi(X) is independent of the environment variables E, they suggest that when E is strongly correlated with Y, then it is possible for Phi obtained via IRM to involve environment variables. They motivate this by suggesting that if there exists a feature mapping Phi(X) = E, it would satisfy the IRM aim, but that this is undesirable.

They suggest instead requiring Phi(X)|Y being invariant to the environment.

The reviewers bring up a couple of concerns. The first is that it is not clear outside some simple examples when Y given Phi(X) being independent of E does not suffice. The second is that the authors also do not empirically validate their fix outside a single simple dataset. Moreover, what are the pitfalls of having Phi(X) given Y being independent of E?

Overall, this is an interesting kernel of an idea; it just needs to be fleshed out a bit more.